



# On the fingerprint of the Antarctica ozone hole in ice core nitrate isotopes: a case study based on a South Pole ice core

Yanzhi Cao[1], Zhuang Jiang[1], Becky Alexander[2], Jihong Cole-Dai[3], Joel Savarino[4], Joseph Erbland[4], Lei Geng[1,5,6]

5    [1]Anhui Province Key Laboratory of Polar Environment and Global Change, School of Earth and Space Sciences, University of Science and Technology of China, Hefei, Anhui, China
[2]Department of Atmospheric Sciences, University of Washington, Seattle, WA, USA
[3]Department of Chemistry and Biochemistry, South Dakota State University, Brookings, SD USA
[4]Univ. Grenoble Alpes, CNRS, IRD, G-INP, Institut des Géosciences de l'Environnement, Grenoble, France
10    [5]CAS Center for Excellence in Comparative Planetology, University of Science and Technology of China, Hefei, Anhui, China
[6]Qingdao National Laboratory for Marine Science and Technology, Qingdao, China

*Corresponding author*: Lei Geng (genglei@ustc.edu.cn)



**Abstract.** Column ozone variability has important implications for surface photochemistry and climate. Ice-core nitrate isotopes are suspected to be influenced by column ozone variability and $\delta^{15}N(NO_3^-)$ has been sought to serve as a proxy of column ozone variability. In this study, we examined the ability of ice-core nitrate isotopes to reflect column ozone variability by measuring $\delta^{15}N(NO_3^-)$ and $\Delta^{17}O(NO_3^-)$ in a shallow ice core drilled at the South Pole. The ice core covers the period of 1944 to 2005, and during this period $\delta^{15}N(NO_3^-)$ was of large annual variability ($(59.2 \pm 29.3)$ ‰), but with no apparent response to the Antarctic ozone hole. Utilizing a snow photochemical model, we estimated 6.9 ‰ additional enrichments in $\delta^{15}N(NO_3^-)$ could be caused by the development of the ozone hole. But this enrichment is nevertheless small and masked by the effects of snow accumulation rate variability in addition to that of the slightly increased snow accumulation rate at the South Pole over the same period of the ozone hole. The $\Delta^{17}O(NO_3^-)$ record displays a decreasing trend by $\sim 3.4$ ‰ since 1976. This magnitude of change can't be caused by enhanced post-depositional processing owing to the ozone hole. Instead, the $\Delta^{17}O(NO_3^-)$ decrease was more likely due to the proposed decreases in $O_3/HO_x$ ratio in the extratropical Southern Hemisphere. Our results suggest ice-core $\delta^{15}N(NO_3^-)$ is more sensitive to snow accumulation rate than to column ozone, but at sites with relatively constant snow accumulation rate, information of column ozone variability embedded in $\delta^{15}N(NO_3^-)$ should be retrievable. In comparison with the South Pole, up to 21 ‰ additional $\delta^{15}N(NO_3^-)$ enrichments can be caused by the ozone hole at Dome A and the signal would be possibly detected if where snow accumulation rate has stayed relatively constant.

## 1 Introduction

Stratospheric ozone absorbs UV radiation protecting life on land. In 1985, Farman et al. (1985) and Stolarski et al. (1986) discovered severe depletion in stratospheric ozone over Antarctica in austral springs since the late 1970s, and this phenomenon was termed the Antarctic ozone hole. The $O_3$ hole in Antarctic spring continued to exist after $\sim$ 1980 (until today) (Müller et al., 2008; WMO, 2018). The Antarctic ozone hole is caused by manmade chlorofluorocarbons (CFCs) which can be photolyzed to HCl and $ClONO_2$ in the stratosphere, producing reactive chlorine and bromine atoms that destroy ozone in catalytic cycles in the cold Antarctic vortex (Keeble et al., 2014; Grooß et al., 2011; Wohltmann et al., 2017). The occurrence of an ozone hole endangers human health and affects ecosystems (Andrady et al., 2017). In addition, the occurrence of an ozone hole also has climate impacts. As suggested by numerous studies, the Antarctic ozone hole results in not only polar lower-stratospheric temperature decrease and polar tropopause rise (Son et al., 2009; Polvani et al., 2011), but also poleward shifts of the midlatitude jet (Archer and Caldeira, 2008) and the Hadley cell edge (Hu and Fu, 2007). In particular, Bitz and Polvani (2012) found the Antarctic ozone hole can cause a broad ocean surface warming and substantial Antarctic sea ice loss probably by enhancing and poleward shifting the westerlies (Polvani et al., 2011; Thompson et al., 2011). Due to the influence of stratospheric ozone depletion (or an ozone hole) on the environment and climate, it is of interest to reconstruct past changes in stratospheric ozone and to assess its response and feedback to climate change. For example, it was suggested that in ~17.7 ka BP there was likely a prolonged Antarctic ozone hole event owing to injection of halogen species to the stratosphere by a series of volcanic eruptions, and this ozone hole may have



accelerated the last Antarctic deglaciation through processes similar to the interactions between the modern Antarctic ozone hole and climate (McConnell et al., 2017).

The occurrence of an ozone hole would result in enhanced surface reception of UV radiation, and therefore influence snow and atmospheric chemistry at the surface. It is known that sunlit snow and ice can lead to emissions of a variety of chemicals (e.g., $NO_x$, OH, molecular halogens, etc.) from snow to the air (e.g., Dominé and Shepson, 2002; Grannas et al., 2007). This affects local atmospheric chemistry through altering the oxidation environment

(Thomas et al., 2012; Zatko et al., 2016) leading to detectable signals in snow chemicals and/or isotopes (Erbland et al., 2013; Frey et al., 2009; McConnell et al., 2017; Spolaor et al., 2021). The latter makes it possible to search for ice-core proxies to explore past changes in stratospheric ozone which determines surface UV radiation. Nitrate is one of the snow impurities sensitive to UV radiation (Frey et al., 2009; Erbland et al., 2013), and previous studies suggest the effect of the perturbation of snow photochemistry on atmospheric chemistry is largely related to snow

nitrate photolysis (Dominé and Shepson, 2002; Grannas et al., 2007; Thomas et al., 2012). Under surface conditions, snow nitrate, which is originally deposited $HNO_3$, can be photolyzed at UV range of 280 to 350 nm (Chu and Anastasio, 2003), and the main product $NO_2$ can quickly reach the overlying atmosphere where it is oxidized to $HNO_3$ again (Zatko et al., 2016; Grannas et al., 2007). The re-produced $HNO_3$ would then be either re-deposited to snow surface or transported away from the site of photolysis (Dibb and Fahnestock, 2004; Erbland et al., 2013;

Davis et al., 2004). This nitrate recycling process at the air-snow interface can occur multiple times before $NO_3^-$ is permanently buried below the snow photic zone which is usually 30 to 60 cm deep and below this depth more than 95 % of the radiation is attenuated (Erbland et al., 2015; Zatko et al., 2013).

Snow nitrate photolysis and the recycling of nitrate at the air-snow interface not only changes snow nitrate concentrations, but also alters snow nitrate isotope composition including $\delta^{15}N$, $\delta^{18}O$ and $\Delta^{17}O$ ($\Delta^{17}O = \delta^{18}O - 0.52 \times$

$\delta^{17}O$) (Frey et al., 2009; Jiang et al., 2021; Erbland et al., 2013; McCabe et al., 2005). In particular, the photolysis itself is associated with nitrogen and oxygen isotope fractionation (Erbland et al., 2013). The specific isotope fractionation constant varies with actinic flux and ranges from, e.g., -78.8 ‰ to -47.6 ‰ for $\delta^{15}N$ and - 34.4 ‰ to - 18 ‰ for $\delta^{18}O$ in the East Antarctica Plateau (Frey et al., 2009; Berhanu et al., 2014; Erbland et al., 2015; Shi et al., 2018). The negative fractionation constants mean enrichment in heavier isotopes (e.g., N-15) in nitrate remaining in

snow after photolysis. For $\delta^{18}O$, the enrichment caused by photolysis is relatively small and the final preserved signal is complicated by oxygen isotope exchange in snow grain and further fractionations/alteration during nitrate recycling. Snow nitrate photolysis doesn't directly influence $\Delta^{17}O$ because it is a mass-independent fractionation signal (McCabe et al., 2005). However, when photolysis occurs, the cage effect leads to oxygen isotope exchange between snow and the intermediate photo-product, lowering snow nitrate $\Delta^{17}O$ (Erbland et al., 2013; Frey et al.,

2009). In addition, the reformed nitrate from snow-sourced $NO_x$ possesses $\Delta^{17}O$ reflecting the local oxidation environment, which is usually different from the $\Delta^{17}O$ signals carried by nitrate from long range transport that is the primary source of snow nitrate. Previous studies suggest $\delta^{15}N$ is most sensitive to the degree of post-depositional processing (Frey et al., 2009; Jiang et al., 2021; Winton et al., 2020). This is because the large nitrogen isotope fractionation constant associated with photolysis leads to significant changes in $\delta^{15}N$ of snow nitrate even with small

amounts of nitrate loss from snow  (e.g., Jiang et al., 2021; Erbland et al., 2013).



The degree of photolysis snow nitrate experienced upon archival below the photic zone and the associated isotope effects are mainly determined by three factors including UV radiation, snow accumulation rate, and snow chemical and physical properties (Zatko et al., 2013). Among them, surface UV radiation is determined by total column ozone (TCO) which is largely controlled by the thickness of the stratospheric ozone layer. Jones and Wolff (2003) also suggested that UV radiation has clearly increased nitrate photolysis frequencies in spring/early summer months (particularly in November) during ozone hole period compared to the pre-ozone hole period. Previous studies suggested $\delta^{15}N$ of nitrate preserved in snow and ice cores has the potential to reconstruct past changes in stratospheric ozone given the sensitivity of snow nitrate $\delta^{15}N$ to the photo-driven nitrate recycling and loss at the air-snow interface (Frey et al., 2009; Erbland et al., 2015; Erbland et al., 2013). Although a recent study suggested that snow nitrate $\delta^{15}N$ could be more sensitive to snow accumulation rate than UV level (Winton et al., 2020), over a period with continuous reoccurrence of ozone layer depletion and thus enhanced surface UV levels (e.g., during the Antarctic ozone hole period), snow nitrate $\delta^{15}N$ may show detectable responses and serve as evidence of past stratospheric ozone change. In addition, changes in TCO may also leave signals in snow $\Delta^{17}O(NO_3^-)$. McCabe et al. (2007) found a co-variation in snowpit $\Delta^{17}O(NO_3^-)$ with TCO at the South Pole with a 2.7-year cycle. A similar cycle of snow $\Delta^{17}O(NO_3^-)$ was also observed at Dome C, Antarctica though the cause was not explicitly explored (Frey et al., 2009). These previous studies imply the potential of ice-core nitrate isotopes in reflecting past stratospheric ozone variability.

Ming et al. (2020) has explored the response of ice-core nitrate $\delta^{15}N$ to possible stratospheric ozone depletions caused by the 1257 C.E. Samalas volcano eruption but found no detectable signals. However, whether or not the Samalas eruption resulted in stratospheric ozone depletion is unknown. The modeled ozone depletion by Ming et al. (2020) was highly sensitive to the amount of HCl injection to the stratosphere. What is more, the stratospheric ozone depletion caused by a single volcanic eruption is an episode event and its effects on ice-core nitrate $\delta^{15}N$ could be masked by annual variations in snow accumulation rate as suggested by Ming et al. (2020). By far, the response of snow and/or ice-core nitrate isotopes to prolonged changes in TCO, in particular, to the Antarctic ozone hole, has not been fully explored, and it remains unclear if ice-core nitrate $\delta^{15}N$ and $\Delta^{17}O$ are able to serve as proxies to reconstruct past TCO. In this study, we measured and examined a 60-year nitrate $\delta^{15}N$ and $\Delta^{17}O$ records covering the period of Antarctic ozone hole from a South Pole ice core. Utilizing a snow nitrate photochemical column model (i.e., the TRANSITs model, Erbland et al., 2015), we quantitatively assessed the effects of the Antarctic ozone hole on nitrate isotopes in the record. The study sheds light on the causes of ice-core $\delta^{15}N$ and $\Delta^{17}O$ variability and the relationships with column ozone density, as well as the potential to use ice-core nitrate $\delta^{15}N$ and $\Delta^{17}O$ to reconstruct past changes in stratospheric ozone.

## 2 Methods

### 2.1 The ice core data

In the austral summer of 2004/2005, two ice cores were drilled at a site (89.96°S, 17.67°W) 4.7 km west of the Amundsen-Scott South Pole station. The two ice cores were less than 10 m apart and named as SP04C5 and SP04C6,





respectively. The SP04C5 core were measured for major ions ($Na^+$, $NH_4^+$, $K^+$, $Mg^{2+}$, $Ca^{2+}$, $Cl^-$, $NO_3^-$, $SO_4^{2-}$) in the laboratory at South Dakota State University using a continuous flow analysis-ion chromatography (CFA-IC) system (Cole-Dai et al., 2006) and the data have been reported by Ferris et al. (2011). Based on the annual peaks of $Mg^{2+}$ and $Na^+$ concentrations, the core was dated and annual snow accumulation rate was estimated (Ferris et al., 2011) .

Nitrate isotope analyses were conducted for the SP04C6 core. Given the close proximity of the two cores, we used the ion concentration measurement and dating results of the SP04C5 core to estimate the annual layer thickness and nitrate abundance of the SP04C6 core and prepared annual samples to contain at least 500 nmol nitrate for triplicate isotope analysis. As a result, a total of 62 samples were cut from the top 8.4 meter of the SP04C6 core covering the years from 1944 to 2005. Among these samples, 14 of them contain less than 400 nmol $NO_3^-$ and thus were

combined with the adjacent ones, resulting in 48 samples for isotope analysis. After cutting, the surface of each sample was cleaned with a bandsaw and the cleaned sample was melted in a clean beaker at room temperature. The nitrate in the meltwater was then concentrated using ion-exchange resin (Frey et al., 2009) and measured for nitrogen and oxygen isotopes using the bacterial denitrifier method with a gold tube at the UW IsoLab. The processes were identical to those described in Geng et al. (2014) and (2015), with analytical precision of $\pm$ 0.25 ‰

and $\pm$ 0.57 ‰ for $\delta^{15}N$ and $\Delta^{17}O$, respectively, as indicated by repeated measurements of the international nitrate isotope standards IAEA-NO-3 and USGS35.

### 2.2 The TRANSITS model simulations

As shown in Fig. 1, compared to years without an ozone hole (represented by the case in 1976), in years with an ozone hole (represented by the year of 1993), surface actinic flux was significantly enhanced in the summer half

year especially in spring when the ozone hole was developed. This presumably would enhance the photo-driven post-depositional processing of snow nitrate and leave more or less changes in the isotopes of the preserved nitrate. Here we used the updated version of the TRansfer of Atmospheric Nitrate Stable Isotopes To the Snow (TRANSITS) model (Erbland et al., 2015; Jiang et al., 2021) to explicitly assess the effects of the Antarctic ozone hole on the ice-core nitrate isotope records from South Pole. The model calculated changes in snow nitrate and its isotopes upon

archival due to the photo-driven post-depositional processing. Fig. 2 illustrates photo-driven post-depositional processing of snow nitrate as described in the TRANSITS model. Here we conducted a long-term simulation covering the full record of the ice core from 1944 to 2005. This includes the pre-ozone hole period before 1976, the ozone hole period from 1976 to the mid-1990s, and after that a slow recovery of column ozone was observed (Zambri et al., 2020). Throughout the record, key parameters (i.e., snow accumulation rate, column ozone density

and snow e-folding depth) influencing the degree of post-depositional processing were determined as follows.
The annual snow accumulation rate was obtained based on ion concentration measurements and the field measured density profile of the SP04C5 core as described in Sect. 2.1 and shown in Fig. 3a. TCO density from 1964 to 2005 was obtained from the NOAA ozonesonde dataset (https://www.esrl.noaa.gov/gmd/dv/data.html) and the austral spring (when the ozone hole occurs in a year) average TCO over this period are plotted in Fig. 3b. Satellite and/or

ground-based ozone observations are not available prior to 1964, so in model calculations we just used the averaged TCO during 1964-1975 (i.e., when there was no ozone hole) to represent TCO from 1944 to 1963.





The e-folding depth of actinic flux in snow is mainly determined by snow density, grain size, and snow light absorbing impurities (LAIs, e.g., dust and black carbon (BC)) (Zatko et al., 2013). Here, we assumed these factors are constant from 1944 to 2005 and the e-folding depth was the same throughout the record for simplicity. Snow density profile at the ice-core drilling site was measured by Ferris et al. (2011) in the field. Due to the lack of measurement of specific surface area (SSA) of snow grains at the South Pole, we applied the average vertical SSA profile measured at Dome C by Gallet et al. (2011) to estimate snow grain size. For LAIs, we adopted a BC concentration of 0.26 ng g$^{-1}$ (Casey et al., 2017) and assuming total LAIs is ~ 10 times of BC to account for non-black carbon materials following Zatko et al. (2013). As a result, the e-folding depth of actinic flux at 305 nm was calculated to be 20 cm at the South Pole, shallower than that estimated by Zatko et al. (2013) (~ 30 cm at remote South Pole where snow impurity concentrations are relatively low probably due to the distance and thus less pollution from the station). Note Zatko et al. (2013) applied Fast-J radiative transfer program which does not consider the absorption enhancement parameter and geometric asymmetry factor when calculating actinic flux. This may explain the difference in the calculated e-folding depths.

The first-order rate constant of snow nitrate photolysis in the photic zone (60 cm, 3 times e-folding depth) was calculated by the following equation:

$$J(z) = \int_{280\,nm}^{350\,nm} \Phi(\lambda) * \sigma_{NO_3^-}(\lambda) * I(z,\lambda) * d\lambda \qquad (1)$$

Where $\Phi$ is the quantum yield, $\sigma$ is the absorption cross section and $I$ is the actinic flux of a wavelength at a depth of z in the photic zone. We used the absorption cross section of $^{14}NO_3^-$ and $^{15}NO_3^-$ in Antarctic snow at a given temperature reported by Berhanu et al. (2014). The depth and wavelength dependent $I$ in snow was calculated by the combination of the Troposphere Ultraviolet and Visible (TUV) radiation model (Madronich et al., 1998) and Two-stream Analytical Radiative TransfEr in Snow (TARTES) model (Libois et al., 2013). Here we used a quantum yield ($\Phi$) of 0.021 molecular photon$^{-1}$ to calculate $J$. This value is within the range of 0.003-0.44 molecular photon$^{-1}$ as observed in laboratory experiments (Meusinger et al., 2014), and best reproduced the observed ice-core $\delta^{15}N(NO_3^-)$ values at the South Pole. The nitrogen isotopic fractionation constant $^{15}\varepsilon_{pho}$ during nitrate photolysis was calculated as the ratio of $^{14}NO_3^-$ and $^{15}NO_3^-$ photolysis rate constant in each layer ($^{15}\varepsilon_{pho} = J^{15}/J^{14} - 1$). The calculated $^{15}\varepsilon_{pho}$ varied from -73.9 to -53.9 ‰ under different solar zenith angles in the summer half year at the South Pole.

Other parameters, for example, the surface meteorological conditions (e.g., boundary layer height, temperature, pressure, etc.), and atmospheric chemical properties including ozone and radical (e.g., OH, RO$_2$ and HO$_2$) concentrations are also needed to calculate the recycling of nitrate in the overlying atmosphere. In the model, the weekly air temperature and surface pressure were obtained from Amundsen-Scott South Pole Station Meteorological Observations dataset (http://amrc.ssec.wisc.edu/usap/southpole/), and the boundary layer height was set as 81 m which is the mean observed value during November and December in 2003 (Neff et al., 2018). Surface ozone concentrations from 1975 to 2005 were obtained from the NOAA ozonesonde dataset (https://www.esrl.noaa.gov/gmd/dv/data.html) and the summer half-year surface O$_3$ concentrations are shown in Fig. 3c. Data prior to 1975 were from data extrapolation. Radicals including OH, HO$_2$, RO$_2$ and BrO in the summer half year were from GEOS-Chem model simulations by Zatko et al. (2016) and assessed using on-site observations by





Kukui et al. (2014) and Mauldin et al. (2004) at the South Pole. The radical time series over the period of the ice core records were then obtained by scaling to local $J_{(NO2)}$ using the relationships between $J_{(NO2)}$ and radicals (Kukui et al., 2014), where local $J_{(NO2)}$ varies with surface actinic flux.

Model inputs of primary nitrate deposition ($F_{pri}$) was set to be $2.4*10^{-6}$ kg N m$^{-2}$ yr$^{-1}$ to best fit the ice core nitrate concentrations at the South Pole after considering post-depositional loss. This value is within the range of 0.09 - $3.5*10^{-6}$ kg N m$^2$ a$^{-1}$ estimated by Zatko et al. (2016) using the GEOS-Chem model. Sources of $F_{pri}$ to the Antarctic atmosphere include stratospheric inputs (FS) and the long-range transport (FT) of nitrate (Erbland et al., 2015; Savarino et al., 2007). Due to the absence of direct observed nitrate fluxes of FS and FT to the South Pole, for simplicity, we assumed FS / $F_{pri}$ = 50 % following Erbland et al. (2015). Note McCabe et al. (2007) estimated that ~25 % of surface snow nitrate at the South Pole is from the stratosphere, but this is an underestimate as in their calculations tropospheric nitrate included snow-sourced nitrate in addition to FT. As estimated by Savarino et al. (2007) and Erbland et al. (2015), $\delta^{15}N(FS)$ and $\Delta^{17}O(FS)$ values were set equal to 19 ‰ and 42 ‰, respectively. Meanwhile, $\delta^{15}N(FT)$ and $\Delta^{17}O(FT)$ were set equal to 0 ‰ and 30 ‰, respectively, following previous Antarctic studies (Morin et al., 2009; Erbland et al., 2015; Winton et al., 2020). The fraction of nitrate export was assumed to be $f_{exp}$ = 20 %, consistent with Erbland et al. (2015) and Winton et al. (2020). The cage effect was as assumed to be 15 % (i.e., the chance of oxygen isotope exchange with water during snow nitrate photolysis) following the methods adopted in previous studies (Winton et al., 2020; Erbland et al., 2015; Jiang et al., 2021). Note we didn't consider their long-term variability for model simplicity. What is more, these are all adjustable starting values justified according either to local observations or previous studies. Starting from these values, the model computes changes of nitrate isotopes upon archival beneath the snow photic zone. Modeled changes in nitrate isotopes after deposition are independent of the starting values but determined entirely by the degree of post-depositional processing experienced before archival (Fig. S1). Model inputs parameters are listed and described in Table S1.

## 3 Results

### 3.1 Ice-core observations

The ice core records cover the years1944 to 2005 C.E. Over this period, annual snow accumulation rate (Fig. 3a) varied significantly from year to year (0.021 - 0.140 w.e. m yr$^{-1}$) (w.e. = water equivalent depth) with a mean of (0.073 ± 0.029) w.e. m yr$^{-1}$. Using on-site stake network measurements at the South Pole station, Mosley-Thompson et al. (1995) found the average snow accumulation rate at the South Pole in the period of 1978 to 1990 was ~0.086 w.e. m yr$^{-1}$, higher than the mean of ~ 0.065 w.e. m yr$^{-1}$ in the 1960s, suggesting a 32% increase. This appears to be consistent with our record from which the mean snow accumulation rate after 1976 (the beginning of the springtime ozone hole) was (0.078 ± 0.026) w.e. m yr$^{-1}$, slightly higher than that before 1970 (0.066 ± 0.030) w.e. m yr$^{-1}$, though the stake measurements indicate a larger increase than our ice core records. The spring TCO (Fig. 3b) record indicates a dramatic decrease from ~300 DU in 1976 to ~150 DU in the mid-1990s, and after that the spring TCO slowly recovered but has not yet returned to the pre-1970 level. The summer half-year average surface $O_3$ concentration record (Fig. 3c) displays a similar trend to spring TCO.





Figure 4a-c show the South Pole ice core records of annual nitrate concentrations ($\omega(NO_3^-)$) and isotopes (i.e., $\delta^{15}N(NO_3^-)$ and $\Delta^{17}O(NO_3^-)$) from 1944 to 2005, respectively. From 2005 to ~2000, there is a decreasing trend in $\omega(NO_3^-)$ and $\Delta^{17}O(NO_3^-)$ while $\delta^{15}N(NO_3^-)$ increases. These are similar to the patterns of snow nitrate concentrations and isotopes versus depth observed at the East Antarctic Plateau (e.g., Morin et al., 2009; Erbland et al., 2013). Throughout the 62-year record, $\omega(NO_3^-)$ ranged from 56.8 to 132.0 ng g$^{-1}$ with an average of ($92.7 \pm 18.5$) ng g$^{-1}$. This is within the range of a recent South Pole ice core of ($58 \sim 169$) ng g$^{-1}$ (Winski et al., 2019) and a 2002 SPRESSO ice core (averaged at $\sim 100$ ng g$^{-1}$) drilled at the South Pole (Jarvis, 2008). All of these ice core results are however lower than $\omega(NO_3^-)$ of (~100 – 200) ng g$^{-1}$ in a 6-m snowpit at the same site reported by McCabe et al. (2007). In comparison, surface snow nitrate concentrations up to 400 ng g$^{-1}$ have been reported by McCabe et al. (2007) and Jarvis (2008).

From 1944 to 2005, $\delta^{15}N(NO_3^-)$ varied from 28.4‰ to 94.9‰ with an average of ($59.2 \pm 29.3$) ‰. This range and the average are also consistent with the SPRESSO ice core ranging from 42 ‰ to 94 ‰ with a mean of ($66.2 \pm 13.0$)‰, except that the very top sample of the SPRESSO ice core possesses $\delta^{15}N(NO_3^-)$ of – 4.4 ‰ (Jarvis, 2008) which is much lower than the very top sample of our ice core of ($33.8 \pm 0.5$) ‰. This is likely due to the fact that our top sample was not actually from surface snow, but covered the first $\sim 40$ cm that represents the weighted mean of snow nitrate in 2004 and part of 2005. In comparison, atmospheric $\delta^{15}N(NO_3^-)$ at the South Pole ranged from - 60.8 ‰ in summer to 10.5 ‰ in winter (Walters et al., 2019). Due to the sparse nature of the atmospheric $\delta^{15}N(NO_3^-)$ data, we can't calculate annual means and compare them with our ice-core data.

From 1944 to 2005, $\Delta^{17}O(NO_3^-)$ varied from 25.7‰ to 34.3‰ with a mean of ($30.0 \pm 1.7$) ‰. Although this range of $\Delta^{17}O(NO_3^-)$ is within that in the 6-m snowpit (20.6 to 33.1 ‰) reported by McCabe et al. (2007), the mean value in our ice core is higher than the mean of ($25.5 \pm 2.1$) ‰ in the snowpit (McCabe et al., 2007). This may be due to spatial variability of the samples but we can't assert the true reason here without more information on sampling and experimental details. Despite this, the ice core record indicates a decrease in $\Delta^{17}O(NO_3^-)$ by $\sim 4$ ‰ from the surface snow to the average below the photic zone in the ice core, while the snowpit record in McCabe et al. (2007) indicates a similar decrease of $\sim 4$ ‰, from 29.3 ‰ in the surface snow to the mean of ($25.4 \pm 1.9$) ‰ below 61 cm depth which is also below the photic zone. These values more or less reflect the degree of $\Delta^{17}O(NO_3^-)$ reduction upon archival at the South Pole. In addition, our ice-core $\Delta^{17}O(NO_3^-)$ record displays a small long-term decreasing trend especially after 1976, and from $\sim 1976$ to 2000 $\Delta^{17}O(NO_3^-)$ decreased by about 3.4 ‰. Atmospheric $\Delta^{17}O(NO_3^-)$ at the South Pole ranged from 21.8 ‰ in summer to 41.1 ‰ in winter (Walters et al., 2019). Unfortunately, the atmospheric data were sparse (only 7 in a year) and one sample represented only a week, so direct comparisons with the ice core annual data cannot be made here.

### 3.2 The modeled results

In Fig. 4, we also plotted the modeled annual $\omega(NO_3^-)$, $\delta^{15}N(NO_3^-)$ and $\Delta^{17}O(NO_3^-)$ from 1944 to 2005. As shown in Fig. 4, the modeled $\omega(NO_3^-)$ and $\delta^{15}N(NO_3^-)$ records capture the long-term variabilities of the observations, including the patterns in the first few years (i.e., from 2005 to ~2000 the $\omega(NO_3^-)$ decrease and $\delta^{15}N(NO_3^-)$ increase). Over the studied period, the modeled average $\omega(NO_3^-)$ and $\delta^{15}N(NO_3^-)$ are ($91.4 \pm 38.1$) ng g$^{-1}$ and ($59.1 \pm 12.8$) ‰,





respectively, similar to the observations. Similar to the observation, the modeled $\delta^{15}N(NO_3^-)$ record also displays no expected response (i.e., increase) in the period of the Antarctic ozone hole (i.e., after ~ 1976). While for $\Delta^{17}O(NO_3^-)$,

the model didn't reproduce the observed decreasing trend from ~1976 to 2000.

In addition, in comparison with the isotope signals of $F_{pri.}$, the preserved $\delta^{15}N(NO_3^-)$ in the record (both observed and modeled) was enriched by ~ 50 ‰. This appears to be smaller than the difference between the observed surface snow $\delta^{15}N(NO_3^-)$ of - 4.4 ‰ (Jarvis, 2008) and snow $\delta^{15}N(NO_3^-)$ of ~ 60 ‰ below the photic zone. This is because surface snow nitrate is affected by snow-sourced nitrate which is depleted in $\delta^{15}N(NO_3^-)$ (as low as – 60 ‰ (Walters

et al., 2019)). While for the modeled $\Delta^{17}O(NO_3^-)$, the ice core mean was reduced by ~ 5.5 ‰ compared to that of $F_{pri.}$, and this level of reduction appears to be larger than the observed ~ 4 ‰ reduction from surface snow to that below the photic zone. Again, this is because surface snow nitrate is influenced by snow-sourced nitrate which possesses lower $\Delta^{17}O(NO_3^-)$ than $F_{pri}$ and is not accounted for in the model.

## 4 Discussion

### 4.1 The effects of post-depositional processing at the South Pole

As shown in Fig. 4, the observed and modeled $\omega(NO_3^-)$ and $\delta^{15}N(NO_3^-)$ exhibits similar trends from 2005 to ~ 2000 (in the top meter of the ice core), i.e., decreases in $\omega(NO_3^-)$ while increases in $\delta^{15}N(NO_3^-)$. Meanwhile, the observed $\Delta^{17}O(NO_3^-)$ also displays a decreased trend which was not pronounced in the model. These features are consistent with the expected effects of post-depositional processing. Summer surface snow $\omega(NO_3^-)$ at the South Pole was

280 reported to be up to ~ 400 ng g$^{-1}$ by McCabe et al. (2007) and Jarvis (2008), this value is about 4 times of the ice core mean $\omega(NO_3^-)$ of (92.7 ± 18.5) ng g$^{-1}$. This difference suggests significant loss of snow nitrate before archival, but the degree of loss cannot be estimated using these number because the summer high $\omega(NO_3^-)$ is due to the re-deposition of snow-sourced nitrate (i.e., the recycling of nitrate at the air-snow interface) in addition to that of primary nitrate.

Regarding the annual nitrate mass loss before archival, the TRANSITs model calculated a mean 42.8 % net loss throughout the record, and the loss was ~ 45 % in the period of the ozone hole which is slightly higher than that of ~ 41% before the ozone hole period. This degree of loss is similar to that (~ 40 %) estimated by Wolff et al. (2002) using a simplified photochemical model at the South Pole. Together with the above estimated level of mass loss, on average the modeled ice-core $\delta^{15}N(NO_3^-)$ was enriched by ~ 50 ‰ compared to $F_{pri.}$ and $\Delta^{17}O(NO_3^-)$ was depleted by

5.5 ‰ before archival. The modeled $\Delta^{17}O(NO_3^-)$ depletion was from two parts: ~ 3.7 ‰ was caused by the reformation of nitrate in the overlying atmosphere which produces nitrate mainly by OH oxidation with relatively low $\Delta^{17}O(NO_3^-)$, and ~ 1.8 ‰ was caused by oxygen isotope exchange with snow grains during photolysis (i.e., the cage effect). In all, the model predicts that at the South Pole, ~ 40 % of nitrate is lost through post-depositional processing inducing significant changes in snow nitrate isotopes compared to atmospheric nitrate.





### 4.2 Effects of the ozone hole on the $\delta^{15}N(NO_3^-)$ and $\Delta^{17}O(NO_3^-)$ records

As discussed in the previous section, snow and ice-core nitrate at the South Pole experiences a significant degree of post-depositional processing that is determined by surface UV radiation (which is in turn influenced by TCO), snow accumulation rate and snow light absorbing impurities. At the South Pole, the TCO record indicates the spring ozone hole started to occur ~ 1976 (Fig. 3b), and repeated every spring with an increasing level of depletion until the mid-1990s. Since the mid-1990s, the ozone hole started to recover but by the end of the record TCO was still lower than that before the 1970s. If other parameters had remained constant, given these changes in the springtime TCO levels, presumably the degree of post-depositional processing should have been enhanced over the period of the ozone hole, and decreases in $\omega(NO_3^-)$ with corresponding increases in $\delta^{15}N(NO_3^-)$ and decreases in $\Delta^{17}O(NO_3^-)$ should be expected. However, as shown in Fig. 4, neither the $\omega(NO_3^-)$ record nor the $\delta^{15}N(NO_3^-)$ record shows expected responses to the ozone hole (shading area in the figure). In comparison, the $\Delta^{17}O(NO_3^-)$ record indicates a decreasing trend starting approximately with the onset of the ozone hole. This appears to be qualitatively consistent with the expected effects of the ozone hole, but the model with consideration of the ozone hole did not reproduce any apparent decreases in $\Delta^{17}O(NO_3^-)$ in the period of the ozone hole.

A recent study at the DML station in Antarctica indicates at this site the preserved snow/ice-core $\delta^{15}N(NO_3^-)$ is more sensitive to snow accumulation rate compared to TCO (Winton et al., 2020). Throughout our record, the mean annual snow accumulation rate during 1976 – 2000 was (0.078± 0.027) w.e. m yr$^{-1}$, slightly higher than that of (0.066 ± 0.030) w.e. m yr$^{-1}$ before 1970. This alone would lead to a mean decrease in the preserved $\delta^{15}N(NO_3^-)$ by ~ 9.0 ‰ as a result of the shortened duration of nitrate in the photic zone, assuming TCO were the same before and after 1976 (Fig.5a, blue dashed line). Meanwhile, if we assumed snow accumulation rate had stayed the same throughout the record, the changes in the springtime TCO after 1976 would lead to a maximum increase in the preserved $\delta^{15}N(NO_3^-)$ by ~ 6.9 ‰ (Fig.5a, green dashed line). These sensitivity tests suggest the overall effect of the ozone hole on preserved $\delta^{15}N(NO_3^-)$ is nevertheless small compared to the observed $\delta^{15}N(NO_3^-)$ values and variability, and this effect has been offset by the effects of the increased snow accumulation rate over the same period (i.e., blue and green lines in Fig. 5 cancel each other after 1976 and make the red line in the model). In addition, in comparison with the maximum level of changes in the preserved $\delta^{15}N(NO_3^-)$ that could be caused by the ozone hole, annual snow accumulation rate variations at the South Pole can lead to ± 14.3 ‰ (1 σ) annual $\delta^{15}N(NO_3^-)$ variability which is also larger than the maximum effects of the ozone hole on preserved $\delta^{15}N(NO_3^-)$. These together mask the effect of the ozone hole although it was gradually enhanced from ~ 1976 to ~ 2000, and explain why the observed and modeled $\delta^{15}N(NO_3^-)$ records display no response to the ozone hole.

In contrast to $\delta^{15}N(NO_3^-)$, the preserved $\Delta^{17}O(NO_3^-)$ is affected by the post-depositional processing mainly through recycling of nitrate in the overlying atmosphere and the cage effect. Over the period of the ozone hole, enhanced surface UV radiation tended to enlarge the cage effect, at the same time, recycled nitrate formed in the overlying atmosphere were also affected by the decreases in surface $O_3$ and increases in $HO_x$ radicals. These changes both tend to lower $\Delta^{17}O(NO_3^-)$. Nevertheless, the model didn't reproduce the observed $\Delta^{17}O(NO_3^-)$ decrease from ~ 1976 to ~ 2000. Even without considering the slightly higher snow accumulation rate in the 1980s and 1990s, the modeled $\Delta^{17}O(NO_3^-)$ in the ozone hole period was only ~ 0.8 ‰ lower than that before 1970 (Fig.5b, green dashed line).





These results suggest that the Antarctica ozone hole has not left any apparent signals in $\Delta^{17}O(NO_3^-)$ either. Note the model didn't consider long term trends in e-folding depth that is determined by snow LAIs (e.g., BC and Dust) because snow LAIs at the South Pole have stayed relatively constant in the past few decades (Winski et al., 2021).

Changes in the export fraction of the snow-sourced nitrate ($f_{exp.}$) would also affect the final preserved $\delta^{15}N(NO_3^-)$ and $\Delta^{17}O(NO_3^-)$, but how this has changed in the past is unknown. The $f_{exp}$ determines how much of the reformed nitrate is recycled back to snow, and the archived $\delta^{15}N(NO_3^-)$ increases with increasing $f_{exp}$, while the archived $\Delta^{17}O(NO_3^-)$ responds oppositely but with a smaller degree of response (Jiang et al., 2021). The fact that only a 3.4 ‰ decrease in $\Delta^{17}O(NO_3^-)$ but no changes in $\delta^{15}N(NO_3^-)$ was observed excludes the possible effects of a varying $f_{exp}$ on the

observed decreasing of $\Delta^{17}O(NO_3^-)$.

In all, the above analysis indicates that the Antarctica ozone hole has not left detectable signals in ice-core $\delta^{15}N(NO_3^-)$ and $\Delta^{17}O(NO_3^-)$ at the South Pole. For $\delta^{15}N(NO_3^-)$, this is mainly because the signal was offset by changes in snow accumulation rate occurring at the same time, in addition to the fact that the ozone hole effects on $\delta^{15}N(NO_3^-)$ is overall small. For $\Delta^{17}O(NO_3^-)$, the effects of the ozone hole were nevertheless small, i.e., only ~ 0.8 ‰

additional $\Delta^{17}O(NO_3^-)$ reduction in the model even without considering the consequences of the changing snow accumulation rate.

In order to search for signals of the ozone hole, we further explored the maximum possible responses of ice-core preserved $\delta^{15}N(NO_3^-)$ to changes in the degree of post-depositional processing caused by the ozone hole at other Antarctic sites. At sites with relatively low snow accumulation rate, the duration of nitrate staying in the photic zone

is longer and thus the effects of ozone hole would be larger. Dome A, East Antarctica has extremely low annual snow accumulation rate (on average ~ 0.023 w.e. m yr$^{-1}$ (Shi et al., 2018)), and using TRANSITS model we estimated that at Dome A there the ozone hole can cause up to ~ 21 ‰ additional enrichments in $\delta^{15}N(NO_3^-)$ in the absence of changes in other factors such as snow accumulation rate or LAIs (SI). Since the effects of the ozone hole were gradually increased given the enhanced level of depletion from ~ 1976 to the mid-1990s, a gradual 21 ‰

increase in $\delta^{15}N(NO_3^-)$ might be possibly detected as long as snow accumulation rate at the Dome A stayed relatively constant before and in the period of the ozone hole. This can be investigated if the Shi et al. (2015) snowpit was dated or new snowpit or shallow ice core from Dome A is measured and dated.

### 4.3 Causes of the observed deceases in $\Delta^{17}O(NO_3^-)$

No apparent changes in ice-core $\omega(NO_3^-)$ and $\delta^{15}N(NO_3^-)$ likely reflect that main nitrate sources to the South Pole

and post depositional effects have not changed in the studied period. The observed decreases in $\Delta^{17}O(NO_3^-)$ thus should be caused by non-local processes other than local snow and atmospheric chemistry. No local processes are related to nitrate chemistry that controls $\Delta^{17}O(NO_3^-)$ in the source regions, as well as atmospheric transport of nitrate to the South Pole. In the discussion as follows, we discern possible reasons responsible for the observed deceases in $\Delta^{17}O(NO_3^-)$ with respect to the possible effects of nitrate chemistry in source regions and atmospheric transport.



### 4.3.1 Effects of oxidant changes in the source regions

Similar decreases in $\Delta^{17}O(NO_3^-)$ over the past few decades were also observed in another Antarctic ice core. Sofen et al. (2014) found that in the WAIS Divide ice core, $\Delta^{17}O(NO_3^-)$ has a long-term downward trend in the past 150 years, and a step decrease occurred after the 1970s. Meanwhile, $\delta^{15}N(NO_3^-)$ in the WAIS Divide ice core over the same period of $\Delta^{17}O(NO_3^-)$ decrease didn't have any long-term trends. This is similar to the South Pole ice core records of $\delta^{15}N(NO_3^-)$ and $\Delta^{17}O(NO_3^-)$. Assisted by box-model sensitivity studies, Sofen et al. (2014) attributed the WAIS Divide ice-core $\Delta^{17}O(NO_3^-)$ decrease in the past 150 years (including that after the 1970s) to decreases in the $O_3$ to $RO_2$ ratio in extratropical Southern Hemisphere $NO_x$ source regions. Decreases in $O_3$ to $RO_2$ ratio means a reduced importance of $O_3$ oxidation in the conversion of NO to $NO_2$, leading to lower $\Delta^{17}O(NO_2)$ and subsequently lower $\Delta^{17}O(NO_3^-)$. Long-range transport of nitrate from the $NO_x$ source regions to Antarctica can then lead to lower $\Delta^{17}O(NO_3^-)$ in primary nitrate. This at least qualitatively explains the observed decreasing $\Delta^{17}O(NO_3^-)$ trend.

Using a global chemical transport model (the GEOS-chem model) with updated sea salt production mechanisms and nitrate chemistry, Alexander et al. (2020) calculated that ~ 80 % of atmospheric nitrate in the Southern Ocean is formed through BrO oxidation pathways. BrO oxidation in general gives higher $\Delta^{17}O(NO_3^-)$ than nitrate formed in other pathways (Morin et al., 2007; Morin et al., 2008; Alexander et al., 2020). Had the relative contribution of BrO chemistry to nitrate formation decreased in the past few decades, atmospheric $\Delta^{17}O(NO_3^-)$ in the Southern Ocean would decrease, and this would also be qualitatively consistent with the ice core records. However, the main source of BrO is acid-catalyzed heterogeneous reactions on sea-salt aerosol (Yang et al., 2008). Sea salt aerosol has not changed significantly in the past few decades as indicated by Antarctic ice-core $Na^+$ records (Cole-Dai et al., 2021; Winski et al., 2021; Ferris et al., 2011), and using ice core ionic balance calculations we calculated $H^+$ concentrations and found acidity has also stayed relatively constant over the period of the record. In addition, sea ice is an important source of sea-salt aerosol (Yang et al., 2008; Huang and Jaegle, 2017), while satellite observations indicate that Antarctic sea ice extent has slightly increased since the 1970s until 2014 (Parkinson, 2019). This tends to enhance BrO production, leading to more nitrate production with higher $\Delta^{17}O(NO_3^-)$, opposite to the observations. Other processes influencing nitrate formation (e.g., daytime vs. nighttime nitrate formation) in the source regions may also have changed and resulted in changes in $\Delta^{17}O(NO_3^-)$. These can be investigated once more nitrate observations in the source regions are available.

### 4.3.2 Effects of primary nitrate inputs

Stratospheric and tropospheric circulation patterns in/around Antarctica in the past few decades have both changed due to the effects of greenhouse warming and/or the ozone hole (Butchart, 2014; Bitz and Polvani, 2012). This would have altered the relative contributions of stratospheric and tropospheric nitrate to the South Pole, with consequences for $\Delta^{17}O(NO_3^-)$ of primary nitrate and thus the ice core $\Delta^{17}O(NO_3^-)$.

Stratospheric nitrate in general possesses higher $\Delta^{17}O(NO_3^-)$ than tropospheric nitrate (McCabe et al., 2007; Savarino et al., 2007). Therefore, reduced stratospheric nitrate input to the South Pole tends to lower $\Delta^{17}O(NO_3^-)$. Stratospheric nitrate input is related to the formation and deposition of polar stratospheric clouds (PSCs) (Jacob, 1999). In the lower part of the stratosphere, polar stratospheric clouds (PSCs) are formed in Antarctic winter when





stratospheric temperature falls below 197 K. PSCs containing nitric acid trihydrate (NAT) can grow sufficiently large and fall out of the stratosphere and deposit on snow surface (Jacob, 1999). This is the so-called stratosphere denitrification, and which can lead to a late winter/early spring nitrate concentration maximum in Antarctic snow as observed (Mulvaney and Wolff, 1993). Observations of abnormally high values of atmospheric $\Delta^{17}O(NO_3^-)$ in

winter/spring across many sites of Antarctica provide further evidence of direct stratospheric nitrate inputs, including the coastal Dumont d'Urville station (Savarino et al., 2007), the inland Dome C station (Erbland et al., 2013), as well as the South Pole station (McCabe et al., 2007). Stratospheric denitrification is directly related to the formation and deposition of PSCs, which in turn is largely determined by lower stratospheric temperature. With the development of the ozone hole, the South Hemisphere lower polar stratospheric temperature has slightly decreased

(Keeble et al., 2014; Polvani et al., 2011), which tends to accelerate polar vortex formation and delays its breakdown, facilitating the growth of PSCs and enhancing stratospheric denitrification. As such, during the period of the ozone hole, $\Delta^{17}O(NO_3^-)$ of stratospheric nitrate inputs to the South Pole should have increased, increasing ice-core $\Delta^{17}O(NO_3^-)$. This is opposite to the observations.

Given the potential increases in the input of stratospheric nitrate, simultaneous increases in tropospheric nitrate

through long-range transport to the South Pole could also lead to $\Delta^{17}O(NO_3^-)$ decreases as long as increases in tropospheric nitrate were more than that in stratospheric nitrate. However, such increases resulting from changes in atmospheric circulation patterns would also influence ice-core concentrations of nitrate as well as other species (e.g., dust) which were not observed in the ice core. Therefore, the observed $\Delta^{17}O(NO_3^-)$ was unlikely caused by changes in the relative importance of stratospheric and tropospheric nitrate to primary nitrate input to the South Pole.

**4.3.3 Effects of interior Antarctic nitrate transport**

Savarino et al. (2007) suggested that snow sourced nitrate from the East Antarctica Plateau can serve as a source of nitrate to other sites downwind the plateau (i.e., transported by the katabatic wind). The South Pole is downwind of the East Antarctic Plateau and influenced by the katabatic winds (Parish and Bromwich, 1991). This means in the summer half year, snow sourced nitrate from the plateau can be transported to the South Pole. The snow sourced

nitrate has low $\delta^{15}N(NO_3^-)$ and $\Delta^{17}O(NO_3^-)$ compared to primary nitrate. For example, at Dome C, the summer atmospheric nitrate which is dominated by snow sourced nitrate possesses $\delta^{15}N(NO_3^-)$ as low as – 40 ‰ that is lower than other nitrate sources, and its $\Delta^{17}O(NO_3^-)$ is < 29 ‰ (Erbland et al., 2013). The latter is similar with the South Pole ice core mean $\Delta^{17}O(NO_3^-)$ but lower than that of primary nitrate to the South Pole. Therefore, increases in transport of snow-sourced nitrate from the plateau to the South Pole would tend to decrease South Pole $\Delta^{17}O(NO_3^-)$

as well as $\delta^{15}N(NO_3^-)$. But a concurrent decrease in $\delta^{15}N(NO_3^-)$ was not observed at the South Pole ice core. What is more, Polvani et al. (2011) suggested that the ozone hole would result in a strengthening of the positive SAM (Southern Annular mode) and surface westerlies in summer, which would further reduce the advections of heat and moisture into East Antarctica (Previdi et al., 2013), weakening the katabatic easterlies over West and East Antarctica (Michiel R. Van Den Broeke and Lipzig, 2003; Marshall and Bracegirdle, 2015). As such, in the period of the ozone

hole, interior Antarctic nitrate transport from the plateau to the South Pole should have been reduced, and which





should lead to increases instead of decreases in South Pole $\Delta^{17}O(NO_3^-)$. Therefore, snow sourced nitrate transported by katabatic wind from the plateau can't explain the observed $\Delta^{17}O(NO_3^-)$ decreases in the South Pole ice core.

**5 Conclusions**

The occurrence of the Antarctic ozone hole has implications for the environment and climate (Bitz and Polvani,
2012; Andrady et al., 2017; Polvani et al., 2011). Reconstruction of past stratospheric ozone or column ozone variability is also important for atmospheric photo-chemistry that influences reactive N, S and other species. To explore the potential of ice-core nitrate isotopes to serve as a proxy for column ozone variability, we measured nitrate isotopes in a 60-year ice core from the South Pole, and used a snow photochemical model to explicitly investigate the effects of the ozone hole on nitrate isotopes in the record. The model results indicate that at the South
Pole post-depositional processing is active, i.e., more than 40% nitrate is lost up on archival, with corresponding enrichments in $\delta^{15}N(NO_3^-)$ by ~ 50 ‰ and reductions in $\Delta^{17}O(NO_3^-)$ by ~ 5.5 ‰. What is more, the model results indicate the ozone hole alone can lead to ~ 6.9 ‰ additional $\delta^{15}N(NO_3^-)$ enrichments compared to the observed average of (59.2 ± 29.3) ‰, and ~ 0.8 ‰ additional reductions in $\Delta^{17}O(NO_3^-)$ compared to the observed average of (30.0 ± 1.7) ‰). These changes are nevertheless small compared to the observed absolute values and uncertainties.
In addition, a slightly increase in snow accumulation rate in the period of ozone hole compared to the pre-ozone hole period also tend to offset the effects of the ozone hole on $\delta^{15}N(NO_3^-)$. As a result, at the South Pole, the ozone hole effects on ice core $\delta^{15}N(NO_3^-)$ is not detectable.

In search of the ozone hole footprints, other sites with lower snow accumulation rate might be better suited. For example, at Dome A, east Antarctica, model results suggest that the ozone hole can cause up to 21 ‰ additional
$\delta^{15}N(NO_3^-)$ enrichments. Had snow accumulation rate at Dome A stayed relatively constant before and in the ozone hole period in the absence of changes in other factors, this gradually increased signal may be detectable. Overall, our analysis suggests ice core $\delta^{15}N(NO_3^-)$ seems to be more sensitive to snow accumulation rate, and detection of column ozone variability from it would require snow accumulation rate being relatively constant or well quantified. Alternatively, if snow accumulation rate is constrained and obtained by other proxies/methods, column ozone
information embedded in ice-core $\delta^{15}N(NO_3^-)$ could be retrieved at least for some recurrent and extreme depletion events.

In contrast to the relatively constant $\delta^{15}N(NO_3^-)$ record, the South Pole ice-core record of $\Delta^{17}O(NO_3^-)$ displays a general decreasing trend especially after the 1970s. Such decreases in the same period have also been observed in the WAIS Divide ice core (Sofen et al., 2014). This decrease can't be explained by post-depositional processing
even including the effects of the ozone hole. Although here we didn't quantitatively discern the causes of the decreasing $\Delta^{17}O(NO_3^-)$, we found this decrease is likely reflecting changes in atmospheric oxidants (i.e., decreases in $O_3$ to $RO_2$ ratio) after the 1970s in extratropical Southern Hemisphere $NO_x$ source regions as suggested by Sofen et al. (2014). Future work would be necessary to explore the recent changes in atmospheric oxidation environment in extratropical Southern Hemisphere as well as the causes and consequences.



**Data Availability**

The meteorological data from Amundsen-Scott South Pole Station Meteorological Observations dataset were freely available at  http://amrc.ssec.edu/usap/southpole/. The TCO density and surface ozone concentrations data from the NOAA ozonesonde dataset can be accessed at https://www.esrl.noaa.gov/gmd/dv/data.html. The ice-core ion concentration and isotopes data will be provided upon direct request to the corresponding author. The codes for
the numerical simulations and their analysis will be provided upon direct request to the corresponding author.

**Author contributions**

LG and BA conceived this study, LG performed the measurements. JD drilled the ice core. YC performed the model simulations, analyzed the data with assistance from ZJ and wrote the manuscript with LG. JS and JE developed the model used in this study. All authors contribution to the writing of the manuscript.

**Competing interests**

The authors declare that they have no conflict of interest.

**Acknowledgments**

L.G. acknowledges financial support from the National Natural Science Foundation of China (Awards: 41822605, 41871051 and 41727901), the Fundamental Research Funds for Central Universities, the Strategic Priority Research
Program of Chinese Academy of Sciences (XDB 41000000), and the National Key R&D Program of China (2019YFC1509100). B.A. acknowledges funding from the US NSF OPP 1446904 and 1542723. J. Cole-Dai thanks the US NSF for funding (Awards 1443663 and 1904142). J.S. was supported by the French National program LEFE (Les Enveloppes Fluides et l'Environnement); the Agence Nationale de la Recherche (ANR) via contract ANR-16-CE01-0011-01 EAIIST ; the Foundation BNP-Paribas through its Climate initiative program and by the French
Polar Institute (IPEV) through programs 1177 (CAPOXI 35–75) and 1169 (EAIIST).

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

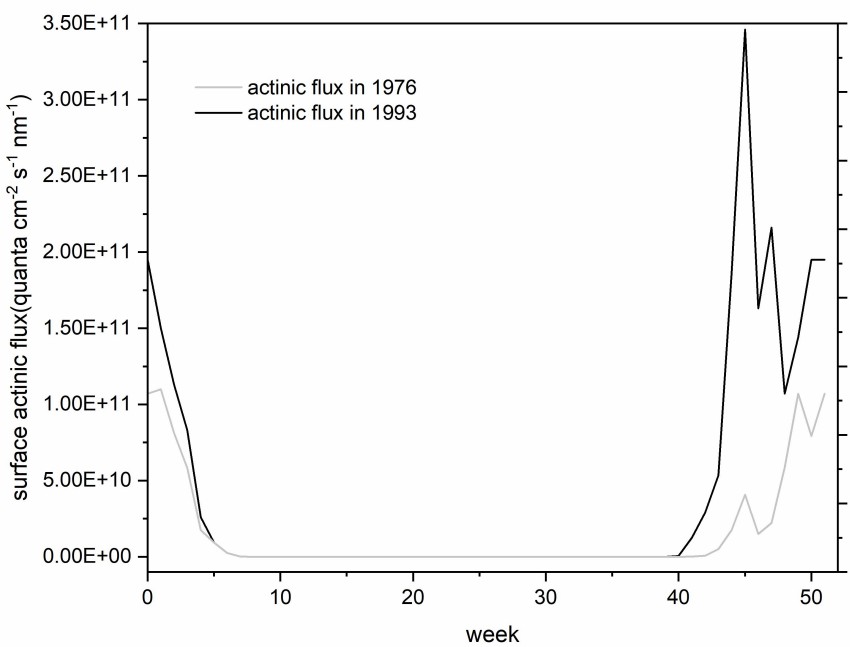


**Figure 1. Surface actinic flux at wavelength of 305 nm (the peak wavelength of snow nitrate absorption spectra at the surface) in pre-ozone hole time (1976) and maximum ozone hole year (1993) at the South Pole. Results were calculated from the TUV radiation model.**





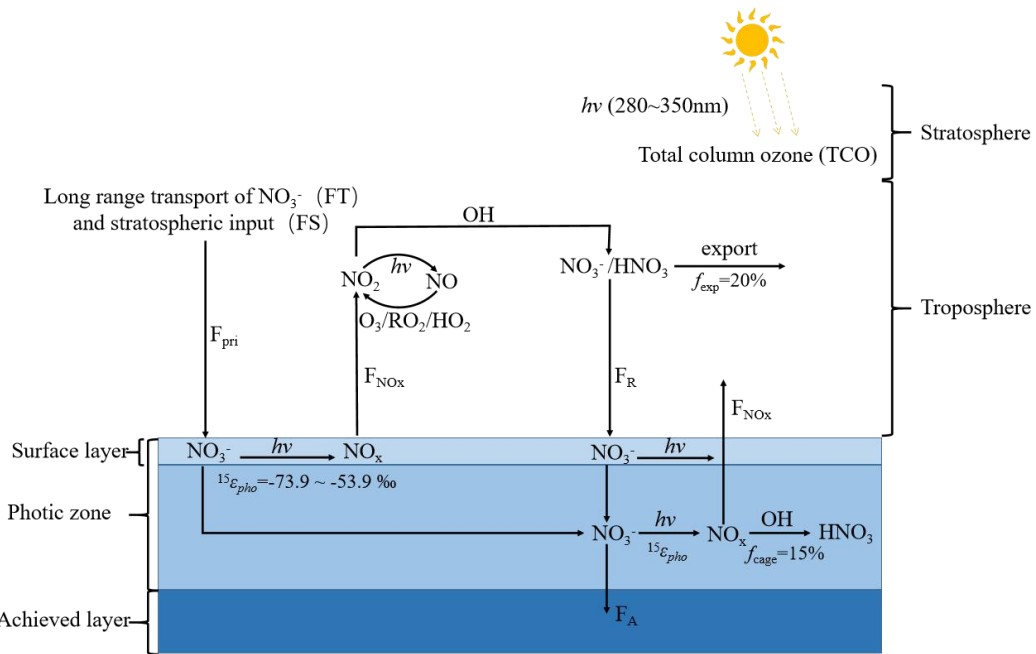

Figure 2. Schematic diagram of the photo-driven post-depositional processing of snow nitrate (adapted from Zatko et al. (2016) and Geng (2022)). Details on abbr. can be found in Table S1.


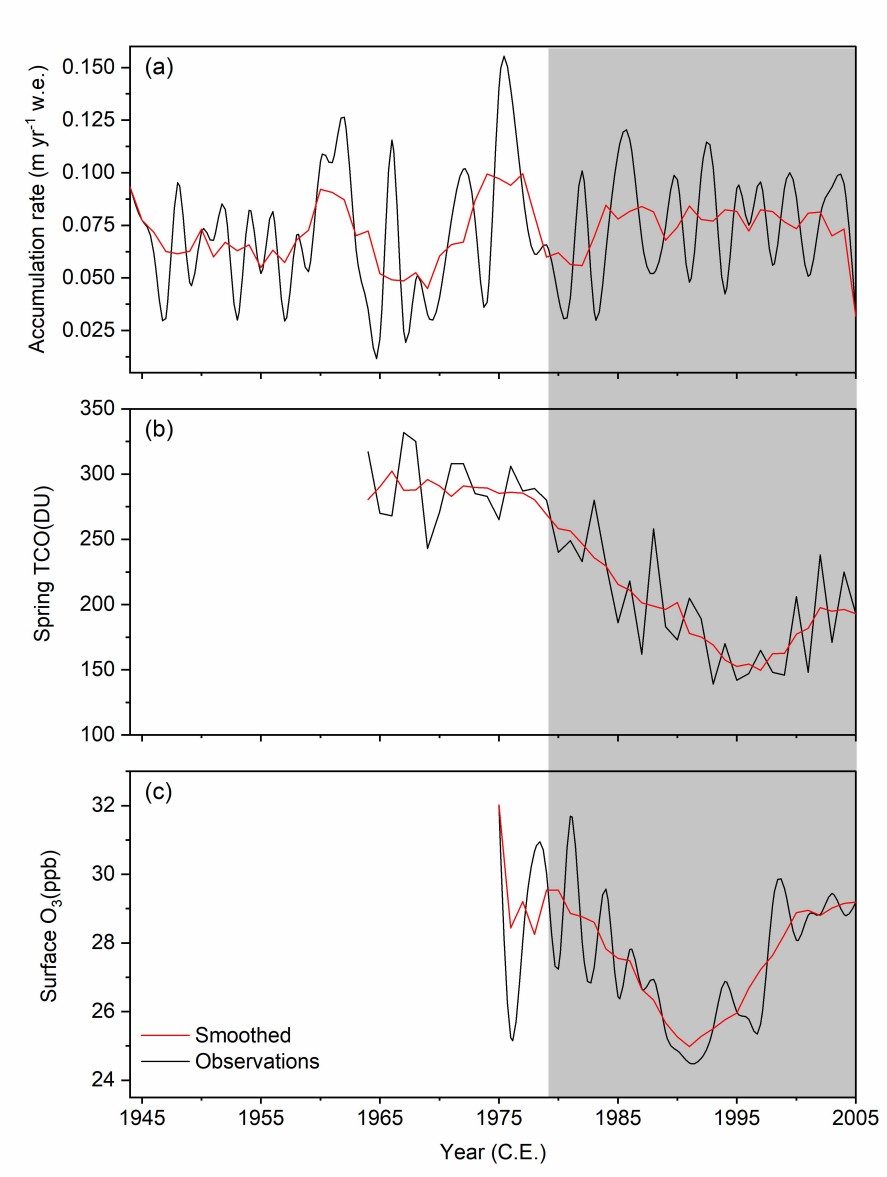

**Figure 3. Time series of annual snow accumulation rate (a), spring (average from September 22 to October 13) TCO (total column ozone) (b), and summer half year surface O₃ concentrations (c) at the South Pole over the period of the ice core record. Red curves are the 5-year moving averages. Grey shading area represents the ozone hole period.**


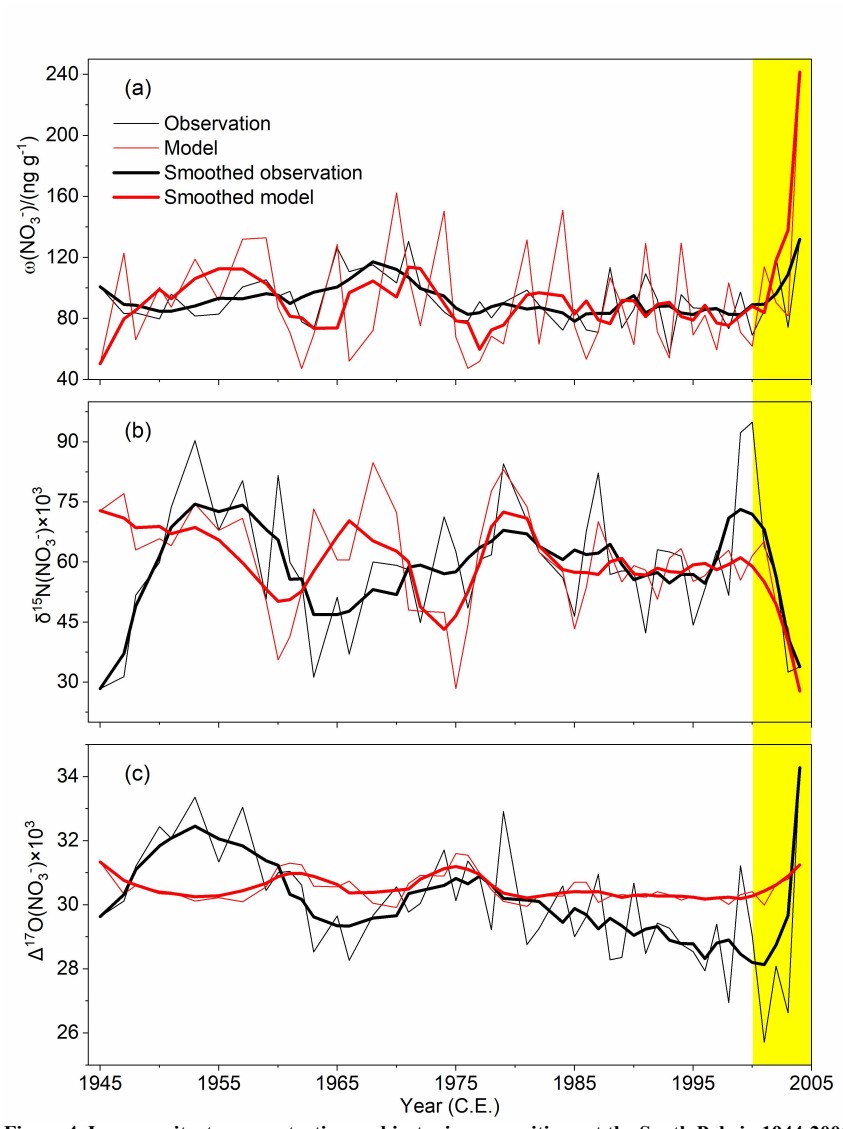

**Figure 4. Ice core nitrate concentration and isotopic compositions at the South Pole in 1944-2005 (black: observations; red: modeled). The thin lines represent the observed and modeled annual (a) $\omega(NO_3^-)$, (b) $\delta^{15}N(NO_3^-)$ and (c) $\Delta^{17}O(NO_3^-)$ from 1944-2005. The thick lines represent the 5-year moving averages. Yellow shaded area represents the period with changes in nitrate concentrations and isotopes from surface snow to below the photic zone.**




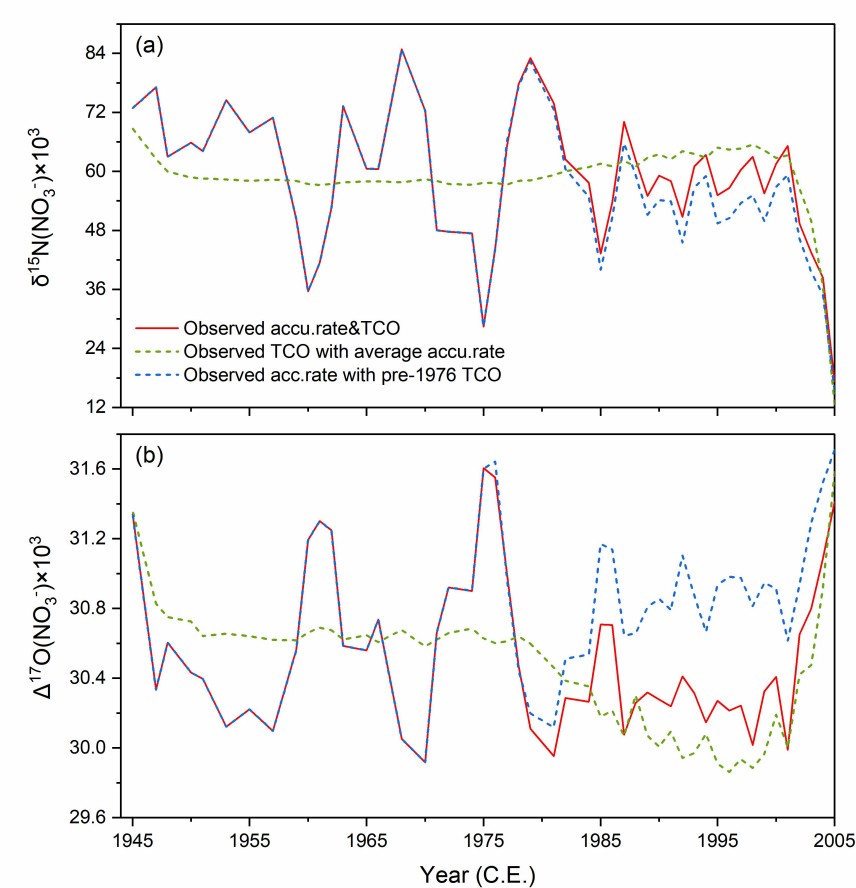

**Figure 5. Sensitivity results of the modeled isotopes, i.e., $\delta^{15}N(NO_3^-)$ (a) and $\Delta^{17}O(NO_3^-)$ (b), to TCO and snow accumulation rate. Red curve: modeled results with observed accumulation rate and TCO; Green curve: modeled results with observed TCO but mean accumulation rate throughout the record; Blue curve: modeled results with observed accumulation rate but TCO were kept the same before and after 1976.**
