# Peer review of "On the potential fingerprint of the Antarctica ozone hole in ice core nitrate isotopes: a case study based on a South Pole ice core"

_Atmospheric Chemistry and Physics, 2022_

## Author Comment (AC2)

We are very grateful to the reviewer for reviewing this manuscript. We have carefully considered the suggestions and made revisions accordingly. Below we list detailed responses to the suggestions and comments. The suggestions and comments are in italics, followed by the response in normal font with changes highlighted in blue.

Cao et al. measure nitrate isotopes and concentrations in a 60-year firn core from South Pole and perform air-snow nitrate transfer simulations using the TRANSITS model to investigate whether nitrate isotopes at the site reflect changes in stratospheric ozone. The results are similar to previous Antarctic studies of ice core with similar snow accumulation rates that indicate d15N(NO3-) is insensitive to total column ozone. Decreases in the D17O(NO3-) record during the ozone hole are qualitatively attributed to atmospheric oxidization changes in the extratropical Southern Hemisphere nitrate source regions. The new dataset is a valuable contribution however, the manuscript could be improved by furthering our understanding of ice core nitrate isotopes in Antarctica which have a unique and not fully understood fingerprint. As such, I believe the authors have an opportunity to use the ice core dataset and the TRANSITS model to advance our understanding of ice core D17O(NO3-) to make a new and valuable contribution to the literature. I look forward to seeing the published.

**Suggestions for improvement**

A paper on nitrate isotopes in a snow pit (1960-2000) from the low-accumulation Dome A site was just published in June (Shi et al., 2022) and the authors conclude that nitrate isotopes (d180, D170, and d15N) record stratospheric ozone depletion and ultra-violet radiation at the Dome A site. The authors have discussed the modelled response of d15N(NO3-) to total column ozone at South Pole versus Dome A sites. Please update the manuscript in light of the newly published paper.

Shi, G., Hu, Y., Ma, H., Jiang, S., Chen, Z., Hu, Z., et al. (2022). Snow nitrate isotopes in central Antarctica record the prolonged period of stratospheric ozone depletion from ~1960 to 2000. Geophysical Research Letters, 49, e2022GL098986. https://doi.org/10.1029/2022GL098986.

**Response:** Thanks for this and we have noted this paper in GRL. The Shi et al. (2022) paper observed changes in nitrate isotopes and concentrations in recent decades and concluded these are due to the effects of the ozone hole. However, carefully reviewing the figures and original data we found where the onsets of the isotope changes are not lined up with the onset of the ozone hole (neither the recovery of ozone hole and the corresponding changes). In addition, they also used the TRANSITs model to estimate changes in isotopes caused by the ozone hole, but the model parameters are not clear, e.g., snow e-folding depth, quantum yield of snow nitrate photolysis, using "similar parameters to Dome C as Erbland et al. (2015)" is impossible to get the reported model results in Shi et al. (2022). For example, "**similar parameters to Dome C as Erbland et al. (2015)**" as stated in Shi et al. 2022 will give a quantum yield of 0.026, applying this value to Dome A would lead to a modeled  $\delta^{15}N(NO_3^-)$  value of ~ 1150 ‰, more than 3 times of the observed value of ~ 300 ‰. What is more, our calculation indicated at Dome A the ozone hole can lead to changes in  $\Delta^{17}O(NO_3^-)$  by ~ 1 per mil the most, while the observed changes in  $\Delta^{17}O(NO_3^-)$  at Dome A is almost 5 per mil. In the revised manuscript we have

updated relevant contents according to Shi et al. (2022) and details can be found later in this response.

Now that there are a number of d15N(NO3-) measurements across Antarctica, a discussion on the sensitivity of d15N(NO3-) and D17O(NO3-) to total column ozone at various ice cores sites, including the new Dome A record, would be valuable addition for the community to make progress on the use of d15N(NO3-) and D17O(NO3-) as a UV or total column ozone proxy.

**Response:** Thanks for this comment. Using TRANSITS model, we calculated the enrichments in  $\delta^{15}N(NO_3^{-})$  and  $\Delta^{17}O(NO_3^{-})$  caused by ozone hole alone (keeping other factors the same) in other East Antarctic sites including Dome A, Dome C, Vostok, Dome Fuji, and a west Antarctic site WAIS Divide in addition to the South Pole. In the revised manuscript, we added a new subsection (new subsection 4.3) to present this table with relevant discussion.

[revised manuscript text omitted]

Another recently published study (July 2022) on nitrate isotopes in relatively high accumulation rate sites (Summit Greenland) also highlights the importance of understanding post-depositional effects of ice core nitrate and it would be worth citing this paper.

Jiang, Z., Savarino, J., Alexander, B., Erbland, J., Jaffrezo, J.-L., and Geng, L.: Impacts of post-depositional processing on nitrate isotopes in the snow and the overlying atmosphere at Summit, Greenland, The Cryosphere, 16, 2709–2724, https://doi.org/10.5194/tc-16-2709-2022, 2022.

**Response:** Thanks for this suggestion. We have added Jiang et al. (2022) as references in the Introduction in the revised manuscript.

There are extremely scare measurements of e-folding depth in Antarctica. A much shallower e-folding depth of 2-5 cm was observed at DML. This was also shallower than estimated by Zatko et al. (2013). What is the uncertainty on your estimated e-folding depth of 20 cm? How appropriate is that estimate in the context of measurements and modelled estimates? Given that recent studies have shown the importance of e-folding depth on nitrate recycling, a discussion and sensitivity analysis of a range of possible e-folding depths for South Pole site is highly encouraged.

**Response:** Thanks for these suggestions. The snow e-folding depth is determined by snow chemical and physical properties. At the South Pole, snow black carbon (BC) concentration measured by Casey et al. (2017) at South Pole is  $(0.26 \pm 0.13)$  ng g-1 (1 $\sigma$ ), the uncertainty on model estimated e-folding depth is about  $\pm$  5 cm (1 $\sigma$ ). Due to the lack of measurement of specific surface area (SSA) and impurities of snow grains at the South Pole, we assumed total LAIs is ~ 10 times of BC and applied the average vertical SSA profile measured at Dome C by Gallet et al. (2011) when calculating e-folding depth at the South Pole.

In addition, we agree that e-folding depth are important to the modeled isotopes. However, here we are focusing on the long-term trends of the isotopes, over the studied period we don't expect e-folding depth have decreased or increased significantly (but of course with annual variabilities). As discussed,

e-folding depth is determined by snow LAIs, and snow physical properties (e.g., density, grain size). Over the studied period, snow LAIs are relatively constant as indicated by South Pole ice core records (as mentioned in the original manuscript). Regarding snow physical properties, the grain size is inversely proportional to specific surface area (SSA) (Zatko et al., 2013), and both snow density and SSA are affected by wind speed and temperature (Kaspers et al., 2004; Domine et al., 2009). Although surface temperature at the South Pole were not changed significantly after the 1950s (below Figure S2), surface wind speed (below Figure S3) displays a decreasing trend since ~ 1970. In Polar Regions the wind action will increase the SSA of surface snow, however, it is the wind storm (>55km/h) that increase the SSA in Antarctica (Domine et al., 2009). Using the empirical relationship between surface snow density with temperature, wind speed and snow accumulation rate in Kaspers et al. (2004) and the parameters in Sugiyama et al. (2012) as follows:

$$o = 305 + 0.629T + 0.150A + 13.5W$$

Where,  $\rho$  is surface snow density in kg/m3, T is the annual average surface temperature in °C, A is the accumulation rate in m w.e. a-1 and W is the annual wind speed in m s-1 at 10 m above the surface. The calculated surface snow density from 1957 to 2005 at the South Pole is plotted in Figure S4. As shown in the figure, snow density after the 1970s (i.e. the ozone hole period) is ~ 20 kg/m3 lower than before that. This would lead to an increase in e-folding depth by only ~1cm, corresponding to ~1 **‰ changes in** the preserved  $\delta^{15}N(NO_3^-)$ . Thus, the effects of the e-folding depth can be ignored.

In the revised manuscript, we added a quick discussion on this (i.e., sensitivity evaluation) as follows: Page 6, line 168: "... Here, we assumed these factors are constant from 1944 to 2005 and the e-folding depth was the same throughout the record for simplicity, though in this period surface wind speed at the South Pole has a decreasing trend (Fig.S3) which may have affected snow density. However, the caused effects on snow e-folding depth due to changing wind speed is only ~ 1 cm and the resulted difference in  $\delta^{15}N(NO_3^-)$  is only ~1 ‰ and thus can be ignored (SI)...". Where in SI, we added the sensitivity discussion on the effects of the possible long-term changes in e-folding depth.

**Figure S2** The annual mean atmospheric temperature from 1957 to 2005 at the South Pole (Surface station data of annual atmospheric temperature and wind speed in the South Pole (https://ramadda.data.bas.ac.uk/repository/entry/show/?entryid=569d53fb-9b90-47a6-b3ca-26306e696 706).

Figure S3 The annual mean wind speed from 1957 to 2005 at the South Pole

---

## Author Response (AR1)

We are very grateful to the reviewer for reviewing this manuscript. We have carefully considered the suggestions and make changes accordingly. Below we list detailed responses to the suggestions and comments. The suggestions and comments are in italics, followed by our responses in normal font with changes highlighted in blue.

*Stable isotopes of nitrate preserved in ice cores hold the potential to reveal past variability of stratospheric ozone over Antarctica. However, there are many factors affecting ice core nitrate concentration as well as its stable isotopic composition. Efforts to understand those processes and their influence are therefore much needed.*

*In this manuscript, Cao et al. presents such an effort using two shallow ice cores from the South Pole dating from 1944 to 2005. Because the time span of the cores nicely encompasses the period of the Antarctic ozone hole since 1976, the nitrate isotope records within serve as a nice archive to investigate the relative contribution of different factors on the nitrate isotopes. Observationally, the authors find that the d15N of nitrate has large variability and the D17O of nitrate displays a long-term decline (on top of the variability). Aided by a snow photochemical model, they conclude that:*

*(1) Ozone hole—which enhances UV flux arrived at the ice sheet surface—alone cannot account for the large variability of d15N, so accumulation rates must be the dominant factor here.*

*(2) Nonetheless, if snow accumulation rates are somewhat stable, the variability could potentially reflect post-depositional processes driven by UV—and by extension by ozone variability.*

*(3) Finally, the trend in D17O seems to be compatible with a change of atmospheric oxidant ratios in the extratropical southern hemisphere.*

*Overall, this paper is timely and interesting, and falls within the scope of Atmospheric Chemistry and Physics. It is well-written and easy to follow. I enjoy reading it and believe it could be published on ACP after making some minor revisions and adding some clarifying statements. I should say, however, that the photochemical modeling is out of my area of expertise, so I am may not be qualified to assess the robustness of the model. I hope other reviewers could comment on the modeling aspect more authoritatively.*

***General comments:***

*First, in the Introduction (from Line 51 and onward) there seems to be no mention of other attempts to reconstruct ozone and the authors proceed to discuss the principles of stable isotopes of nitrate preserved in ice cores as a potential ozone proxy. Non-ozone specialists may wonder if there are other ways to know ozone in the past. A quick review of the existing methods with their strengths and limitations discussed could be helpful here. The readers will also be able to understand the value of the isotope records in ice-core nitrate.*

**Response:** We agree with the reviewer that it would be great if there are other proxies can be used to reconstruct past column ozone density. But to our best knowledge, UV-sensitive chemical species preserved in snow are the known potential candidates, and nitrate is one of them and is the mostly studied. To make this point more cleared, in the revised manuscript we have added a statement at the beginning of this paragraph as follows:

"However, to reconstruct past changes in stratospheric ozone is difficult due to the lack of reliable proxies. UV-light sensitive chemicals in snow including nitrate (Frey et al., 2009) and bromine (Abbatt et al., 2012) have been sought to investigate changes in surface UV conditions and the potential links to

stratospheric ozone. The occurrence of an ozone hole…"

*Second, in the Discussion 4.3, the lines of reasoning could benefit from a simple restructure: why not putting the 4.3.2 and 4.3.3 first? This way you could discuss the reject the alternative hypotheses, leaving the most plausible explanation (changes in the O3/HOx ratio) on the table.*

**Response:** Thanks for this suggestion. We have changed the orders of the subsections in 4.3 to make the most plausible explanation in the last subsection.

*Third, one key point of the paper is that Dome A might be a good place to study nitrate isotopes because of its low snow accumulation rates. This might foreshadow a follow-up study from that very site, which is great. For the present study, however, can you also calculate the expected d15N variability induced by stratospheric ozone in other East Antarctic sites such as Vostok, Dome C, and Dome F where deep ice cores have been drilled? This could be summarized with a new figure. Though it does not necessarily mean that you have those samples, I think this exercise could benefit the ice core communities in general.*

**Response:** Thanks for this suggestion. Using TRANSITS model, we calculated the enrichments in $\delta^{15}N(NO_3^-)$ and $\Delta^{17}O(NO_3^-)$ caused by ozone hole alone (keeping other factors the same) in other East Antarctic sites including Dome A, Dome C, Vostok, Dome Fuji, and a west Antarctic site WAIS Divide in addition to the South Pole. In the revised manuscript, we added a new subsection (new subsection 4.3) to present this table with relevant discussion.

[revised manuscript text omitted]

*Specific comments:*

*Line 21: "but" and "nevertheless" are repetitive. "Nevertheless, this enrichment is small and masked by ..." sounds better.*

**Response:** Thanks for this suggestion. We have made this correction in the revised manuscript.

*Line 21: the second half of this line could be simplified by saying "... masked by the effects of snow accumulation rates at the South Pole ..." In essence the snow accumulation rates have two parts: internal variability superimposed on a long-term trend. They could be discussed in greater detail in the main text without complicating the message here in the abstract.*

**Response:** Thanks for this suggestion. We have made this correction in the revised manuscript.

*Line 32: consider changing "protecting life on land" into "and protects life on land". No need for using the nonfinite verb here.*

**Response:** Thanks for this suggestion. We have made this correction in the revised manuscript.

*Line 44: missing an "of" after "shifting".*

**Response:** Thanks for this suggestion. We have made this correction in the revised manuscript.

*Line 57: "ozone which determines surface UV radiation." This seems to suggest that there are lots of*

*"ozone" and the one being talked about is the one that determines surface UV radiation. Yet, in fact you are just describing stratospheric ozone, so no need for the defining relative clause here, and there should be a comma "," before "which".*

**Response:** Thanks for this suggestion. We have deleted the words after ozone in this sentence in the revised manuscript

*Line 61: missing an "as" after "deposited".*

**Response:** Thanks for this suggestion. We have made this correction in the revised manuscript.

*Line 78: this sentence is not very clear. By saying "it is a mass-independent fractionation signal" it is implied that photolysis is a mass-dependent process. If this is the case, please explicit state so.*

**Response:** Yes, the photolysis is a mass-dependent process. We add the following statement in our revised text:

"…and further fractionations/alteration during nitrate recycling. Snow nitrate photolysis doesn't directly influence $\Delta 17O$ because it is a mass-independent fractionation signal while photolysis only induces mass-dependent fractionation (McCabe et al., 2005)."

*Line 129: can you specify which years were binned to the adjacent samples? This could be provided as a supplementary table.*

**Response:** Thanks for this suggestion. The samples that were combined with the adjacent ones and the corresponding years were summarized in the table below and the table has been added as SI table in the revised manuscript:

| Sample ID | Corresponding years (C.E.) |
|---|---|
| S1 | 2004-2005 |
| S21 | 1983-1984 |
| S23 | 1980-1981 |
| S29 | 1973-1974 |
| S32 | 1969-1970 |
| S33 | 1967-1968 |
| S35 | 1964-1965 |
| S40 | 1958-1959 |
| S41 | 1956-1957 |
| S42 | 1954-1955 |
| S43 | 1952-1953 |
| S45 | 1949-1950 |
| S47 | 1946-1947 |

| S48 | 1944-1945 |
|------|-----------|

*Line 190: "were from data extrapolation" could be better phrased as "were extrapolated".*

**Response:** Thanks for this suggestion. We have made this correction in the revised manuscript.

*Line 216: missing a blank between "years" and "1944".*

**Response:** Corrected as suggested.

*Line 218, 231, 237, 245, and 251: please specify the meaning of the number after the sign. Is it one standard deviation?*

**Response:** Yes, it is one standard deviation. We have added (1σ) after these values to indicate the meaning of the number after the sign.

*Line 263: there are two "similar to the observation". Please consider rephrasing.*

**Response:** Thank you for raising this issue. We have rewritten the sentence as follows:

"…Over the studied period, the modeled average ω(NO3-) and $\delta^{15}N(NO_3^-)$ are (91.4 ± 38.1) ng g-1 (1σ) and (59.1 ± 12.8) ‰ (1σ), respectively, similar to the observations. The modeled long-term trend in $\delta^{15}N(NO_3^-)$ is also similar to the observation and displays no expected response…".

*Line 278: change "pronounced" to "reproduced"?*

**Response:** Corrected as suggested.

*Line 283: I would appreciate you putting the numbers into a greater perspective here. At face values, about 75% of the primary nitrate was lost, leaving 25% nitrate behind. On the other hand, you mentioned that re-deposited nitrate contributed to the preserved nitrate. Does this mean that the loss of \*primary\* nitrate exceeds 75%? Similarly, please specify what the ~40% nitrate loss calculated by the photochemical model refers to, perhaps with the help of Figure 2: is the nitrate in the combined surface and photic layer?*

**Response:** We guess the face-value of 75 % was estimated by comparison of the **summer surface** concentration (ω(NO3-) = ~ 400 ng g$^{-1}$) and that of the firn core average ~ 100 ng g$^{-1}$. First of all, this is not correct, as one have to use at least the annual mean of the surface snow concentration as the starting point to estimate the lost, and in winter surface snow concentration is much lower than in summer as observed by Walters et al. 2019 at the South Pole (this is also true for other polar sites).

   Regarding the reported ~ 40 % loss from our calculation, this is the net loss, i.e., the difference between the finally preserved nitrate and primary nitrate. In the revised manuscript, we have added Equation.2 to indicate how this is calculated: ($f_{loss}$):

$$f_{loss} = 1 - \frac{F_A}{F_{pri}} \tag{2}$$

where $F_A$ represents the archival flux of nitrate (Fig. 2).

*Line 305: the shading area in Figure 4 does not correspond to the periods with an ozone hole.*

**Response:** In the revised manuscript, we merged Figure 3 and Figure 4 and plotted a new Figure 3. Grey shading area corresponds to the ozone hole period.

[Figure]

**Figure 3.** Left panels: time series of annual snow accumulation rate (a), spring (average from September 22 to October 13) TCO (total column ozone) (b), and summer half year surface $O_3$ concentrations (c) at the South Pole over the period of the ice core record. Red curves are the 5-year moving averages. Right panels: ice core nitrate concentration and isotopic compositions at the South Pole in 1944-2005 (black: observations; red: modeled). The thin lines represent the observed and modeled annual (d) $\omega(NO_3^-)$, (e) $\delta^{15}N(NO_3^-)$ and (f) $\Delta^{17}O(NO_3^-)$ from 1944-2005. The thick lines represent the 5-year moving averages. Yellow shaded area represents the period with changes in nitrate concentrations and isotopes from surface snow to below the photic zone. Grey shading area represents the ozone hole period.

*Line 307: is this from the sensitivity test? If so Figure 5 should be mentioned. Alternatively, you could just discuss d15N of nitrate exclusively in section 4.2 (which now needs a new title of course), and leave the discussion of D17O entirely to the next section.*

**Response:** No, this is from the TRANSITs modeled result shown in Fig.3f above. We have made more detailed illustrations as follows:

"…In comparison, the observed $\Delta^{17}O(NO_3^-)$ record indicates a decreasing trend starting approximately with the onset of the ozone hole (Fig.3f, black lines). This appears to be qualitatively consistent with the expected effects of the ozone hole, but the model with consideration of the ozone hole did not reproduce any apparent decreases in $\Delta^{17}O(NO_3^-)$ in the period of the ozone hole (Fig.3f, red lines).".

*Line 364: "discern" might not be the proper word choice here. "Investigate" or "Examine" sounds more logical.*

**Response:** We agree and have revised it accordingly.

*Line 455: "Had snow accumulation rate at Dome A stayed …" Technically this sentence shouldn't be in subjunctive mood, because by doing so you are implying that accumulation rates at Dome A were, in fact, not stable. Yet, the accumulation rate history is not known, so you could just use "If" instead of "Had" to indicate a possibility.*
**Response:** We agree and have revised it accordingly.

*Figure 2: should be "Archived" instead of "Achieved" layer?*
**Response:** Yes, I have revised it in manuscript.

*Figure 3: per the text, the "ozone hole period" begins right after 1976, but in the figure here, the ozone hole starts around 1979 C.E.? Please make them consistent with each other.*
**Response:** Thanks for this suggestion. We have made this consistent (~ 1976) in the revised manuscript.

*Figure 4: please add some visual guidance to mark the ozone hole period.*
**Response:** We have added grey shading area in the Figure to mark the ozone hole period.

*Figure 5: same as Figure 4, a little visual cue of the ozone hold period (or simply the beginning of it) would be nice.*
**Response:** Thanks for this suggestion. We have done this in the revised manuscript.


$$\rho = 305 + 0.629T + 0.150A + 13.5W$$

Where, $\rho$ is surface snow density in $kg/m^3$, T is the annual average surface temperature in °C, A is the accumulation rate in m w.e. $a^{-1}$ and W is the annual wind speed in m $s^{-1}$ at 10 m above the surface. The calculated surface snow density from 1957 to 2005 at the South Pole is plotted in Figure S4. As shown in the figure, snow density after the 1970s (i.e. the ozone hole period) is ~ 20 $kg/m^3$ lower than before that. This would lead to an increase in e-folding depth by only ~1cm, corresponding to ~1 **‰ changes in** the preserved $\delta^{15}N(NO_3^-)$. Thus, the effects of the e-folding depth can be ignored.

In the revised manuscript, we added a quick discussion on this (i.e., sensitivity evaluation) as follows: Page 6, line 168: "… Here, we assumed these factors are constant from 1944 to 2005 and the e-folding depth was the same throughout the record for simplicity, though in this period surface wind speed at the South Pole has a decreasing trend (Fig.S3) which may have affected snow density. However, the caused effects on snow e-folding depth due to changing wind speed is only ~ 1 cm and the resulted difference in $\delta^{15}N(NO_3^-)$ is only ~1 ‰ and thus can be ignored (SI)...". Where in SI, we added the sensitivity discussion on the effects of the possible long-term changes in e-folding depth.

[Figure]

**Figure S2** The annual mean atmospheric temperature from 1957 to 2005 at the South Pole (Surface station data of annual atmospheric temperature and wind speed in the South Pole (https://ramadda.data.bas.ac.uk/repository/entry/show/?entryid=569d53fb-9b90-47a6-b3ca-26306e696 706).

[Figure]

**Figure S3** The annual mean wind speed from 1957 to 2005 at the South Pole

[Figure]

**Figure S4** The calculated surface snow density from 1957 to 2005 at the South Pole

*Please add a section of assessing the validity of the TRANSITS model especially in regards to D17O(NO3-). The model doesn't simulate the observed decreasing D17O(NO3-) trend from ~1976 to 2000. Why is this? How much can you take away from the simulated D17O(NO3-) results? How can you improve the model? How does the model help you understand D17O(NO3-) at South Pole. TRANSITS simulations of D17O(NO3-) would be an area where the authors can contribute new understanding to the literature.*

**Response:** Thanks for this suggestion. But the TRANSITS model is an air-snow exchange model, it only includes snow chemistry and chemistry in the overlying atmosphere. This is saying, the model can only be used, or the best used, to investigate changes related to what occurs locally (in snow or the above). It is from the model results that we can conclude that the local processes (i.e., largely the post-depositional processing) cannot result in any long-term changes in $\Delta^{17}O(NO_3^-)$, so that the observed decreasing $\Delta^{17}O(NO_3^-)$ should be from other factors which are most likely related to changes in primary nitrate (the starting values of $\Delta^{17}O(NO_3^-)$). From here, we further discussed effects of source regions oxidation environment and varying transport on $\Delta^{17}O(NO_3^-)$ of primary nitrate. To thoughtfully investigate and/or discern the reasons why $\Delta^{17}O(NO_3^-)$ of primary nitrate has decreased since the 1970s, a chemical transport model with detailed NOx source and chemistry changes in the past are necessary. This is however out of the scope of this study but in our to-do-list.

*Introducing the South Pole site in terms of the snow accumulation and also atmospheric nitrate isotopes (Walters et al., 2019) in the introduction would be helpful to put the site into context of other records given that the nitrate isotopes are sensitive to accumulation rate.*

**Response:** Thanks for this suggestion. In the end of the introduction part, we have added the following statements: "…In this study, we have …. from a south Pole ice core. At the South Pole, snow accumulation rate is relatively low (0.073 w.e. m yr$^{-1}$.) and the effects of post-depositional processing are well observed as reflected by the large differences between atmospheric and snow nitrate isotopes (Walters et al. 2019). …"

*It is not always clear in the discussion if the authors are talking about the results from TRANSITS or observations.*

**Response:** Thanks for your comment. In the revised manuscript we have explicitly distinguish the simulated and observed results by adding describing terms of 'observed' or 'modeled' when related content is discussed or mentioned.

***Specific comments***

*L1 The title is misleading as nitrate isotopes at South Pole do not reflect changes stratospheric ozone changes.*

**Response:** Thanks for this suggestion. We have changed the title as "On the potential fingerprint of the Antarctica ozone hole in ice core nitrate isotopes: a case study based on a South Pole ice core".

*L26 HCl and ClONO2*

**Response:** This seems to be in L36, and we have added a word 'produce' before "HCl and ClNO2' if this is the review meant.

*L65-67 The photic zone at DML is 15 cm which is less than Dome C (Winton et al., 2020).*

**Response:** Thanks for your comment. We have revised it as follows:

**"…** This nitrate recycling process at the air-snow interface can occur multiple times before NO3- is permanently buried below the snow photic zone which is usually 15 to 60 cm deep and below this depth more than 95 % of the radiation is attenuated (Erbland et al., 2015; Zatko et al., 2013; Winton et al., 2020)."

*L71-73 This sentence focusses on fractionation constants on the EAP. Relevant to this study are fractionation constants in the "transition zone" characterized by snow accumulation rates typical of sites located between the EAP and coast (5–20 cm yr−1 w.e.; Erbland et al. 2015).*

**Response:** No, here we meant fractionation constant associated with snow nitrate photolysis, which is only related to wavelength. Erbland et al. (2015) derived different fractionation constants at different sites, but which are "apparent fractionation constant" and basically are mixed effects from the whole post-depositional processing (photolysis is the trigger step).

*L86-102 Recent studies have shown the importance of e-folding depth on nitrate recycling. This is important to mention here.*

**Response:** We thank the reviewer for pointing this, but e-folding depth is essentially determined by snow chemical and physical properties which are more direct when discussing post-depositional processing.

*L108-111 See the recently published paper by Shi et al. (2022)*

**Response:** Thanks for this suggestion. We add the following statement in our revised text:

"…annual variations in snow accumulation rate as suggested by Ming et al. (2020). A recently study by Shi et al. (2022) measured firn core nitrate concentrations and isotopes at Dome A, Antarctica, and there appears to be responses of nitrate concentrations and isotopes to the Antarctic ozone hole. However, the onsets of the observed changes (i.e., isotopes and column ozone) are not lined-up, the model efforts in the study is ambiguous and it is unclear whether the effects of the ozone hole can quantitatively (or even at the qualitive level) explain the observed changes. In this study, we measured and examined …"

*L131 Did you decontaminate the samples?*

**Response:** Yes. As stated in lines 137 to 139, we cleaned samples surface with a bandsaw and melted samples in a clean beaker at room temperature, and the concentrated process using ion-exchange resin can also filter meltwater and further decontaminate the samples.

*L134 Suggest moving reference to Geng et al. further up in the methods section.*

**Response:** Thanks for this suggestion. Revised it accordingly

*L119-136 Please add protocols for minimising contamination. Please state the sample resolution in terms of depth and age here.*

**Response:** Thanks for your comment. In the original manuscript we have described the protocols for minimizing contamination. "… After cutting, the surface of each sample was cleaned with a bandsaw and the cleaned sample was melted in a clean beaker at room temperature. The nitrate in the meltwater was then concentrated using ion-exchange resin…". We add the following statement in the Sect.2.1 in our revised text for the sample resolution in terms of depth and age:

"…As a result, a total of 62 samples were cut from the top 8.4 meter of the SP04C6 core covering the years from 1944 to 2005. The depth resolution of these samples varies from 11 to 38 cm and each sample covering 1 year. Among these samples, …"

*L133 UW*

**Response:** Corrected as suggested

*L138-140 This sentence seems out of place.*

**Response:** Thanks for this suggestion. The purpose of this sentence is to show that the ozone hole does cause a significant increase in surface UV radiation. This would further enhance the photo-driven post-depositional processing of snow nitrate and can assess through TRANSITs model. We add the following statement in revised text:

"…As shown in Fig. 1, compared to years without an ozone hole (represented by the case in 1976), in years with an ozone hole (represented by the year of 1993), surface actinic flux was significantly enhanced in the summer half year especially in spring when the ozone hole was developed. The stronger surface actinic flux in the ozone hole period presumably would enhance the photo-driven post-depositional processing…"

*L164 How did you calculate the e-folding depth?*

**Response:** Thanks for your comment. We used the two-stream Analytical Radiative TransfEr in Snow

(TARTES) model (Libois et al., 2013) to calculate the depth profile of actinic flux at different wavelength. We have described this in revised manuscript. We have revised it as follows:

"As a result, the e-folding depth of actinic flux at 305 nm was calculated to be 20 cm at the South Pole using the Two-stream Analytical Radiative TransfEr in Snow (TARTES) model (Libois et al., 2013), shallower than…"

*L223-226 Seems out of place.*

**Response:** Thank you for raising this issue. The observed total column ozone and surface ozone should not be put in Sect.3.1 describing ice-core observations. We have moved these two sentences describing spring TCO and surface ozone trends to Sect.2.2 in revised manuscript where ozone data were first mentioned.

*L234 Add the dates of the pit*

**Response:** Thanks for pointing out this. We have added the snowpit date in revised manuscript:

"…All of these ice core results are however lower than ω(NO3-) of (~100 – 200) ng g-1 in a 6-m snowpit (1977-2003) at the same site reported by McCabe et al. (2007)."

*L282 Can you use the approach of Weller et al. (2004) to calculate nitrate loss? And then compare to the TRANSITS estimate of nitrate loss?*

**Response:** Thanks for your comment. Weller et al. (2004) quantified the total loss of NO3- by comparing snow nitrate concentration of the first year to the 100-year mean concentrations retrieved from the firn core. They used the first-year snow concentration to represent the surface snow concentration. Unfortunately, our ice core was drilled in the winter season of 2004/2005, the first few center meter sample represents the first couple of months in 2005. This means we don't really have the "first year" data/ We want to note, this is not a good approach as even the first year data is available it is already affected by post-depositional processing and not equal to primary nitrate. Weighted monthly average surface snow concentration could be better as the starting point.

*L295 Heading should reflect that this section is about TRANSITS modelling*

**Response:** Thanks for your suggestion. We changed the heading into: "4.2 Modeled effects of the ozone hole on the $\delta^{15}N(NO_3^-)$ and $\Delta^{17}O(NO_3^-)$ records"

*L334 This assumption ignores other factors that influence e-folding depth. While we don't know how e-folding depth changes over time, based on changes in grain size, snow density and impurity content it is fair to assume e-folding depth at any site is not constant through time. Sensitivity studies show that nitrate isotopes are sensitive to changes in e-folding depth.*

**Response:** Thanks for this comment. Ice-core records shows that in the past 50 to 100 years at the South Pole impurity (i.e., LAI) are relative constant (no decreasing or increasing trend). Regarding the physical peripteries, we have assessed them in earlier response and their long-term effects on e-folding and the consequences on $\delta^{15}N(NO_3^-)$ are negligible. In the revised manuscript, we have explicitly discussed this in supplemental materials.

*L346-357 Update in light of the published work by Shi et al. (2022).*

**Response:** Thanks for your suggestion. We have added a subsection to discuss other sites including

Dome A.

[revised manuscript text omitted]

*L359 The ice core data in the figures suggest interannual variability.*

**Response:** Thanks for your suggestion. In this section we discussed the long terms. The revised text is shown as follows:

"…No apparent long-term trends in ice-core $\omega(NO_3^-)$ and $\delta^{15}N(NO_3^-)$ likely reflect that main nitrate sources to the South Pole and post depositional effects have not changed in the studied period…"

*L391 A concluding sentence about oxidation for this paragraph would be helpful here.*

**Response:** Thanks for your suggestion. According to the suggestions of Referee # 1, we have changed the orders of the subsection in the revised manuscript, and in this part, we have added a conclusion sentence as follows:

"…Therefore, the observed $\Delta^{17}O(NO_3^-)$ decrease after the 1970s is more likely due to the potential decreases in $O_3/HO_x$ ratio in the extratropical Southern Hemisphere. This remains to be explored and confirmed with future studies"

*L424 The EAST ANTARCTIC PLATEAU snow sourced*

**Response:** Corrected as suggested.

*Figures: It would be very helpful for the reader to visualise the TCO and nitrate isotope trends on the same figure.*

**Response:** Thanks for your suggestion. In the revised manuscript, we have merged Figure 3 and Figure 4 and plotted a new Figure 3. Grey shading area correspond to the first stage of the ozone hole period (from 1976 to 1996).

[Figure]

**Figure 3.** Left panels: time series of annual snow accumulation rate (a), spring (average from September 22 to October 13) TCO (total column ozone) (b), and summer half year surface $O_3$ concentrations (c) at the South Pole over the period of the ice core record. Red curves are the 5-year moving averages. Right panels: ice core nitrate concentration and isotopic compositions at the South Pole in 1944-2005 (black: observations; red: modeled). The thin lines represent the observed and modeled annual (d) $\omega(NO_3^-)$, (e) $\delta^{15}N(NO_3^-)$ and (f) $\Delta^{17}O(NO_3^-)$ from 1944-2005. The thick lines represent the 5-year moving averages. Yellow shading area represents the period with changes in nitrate concentrations and isotopes from surface snow to below the photic zone. Grey shading area represents the ozone hole period.

*Fig. 5: please add in the nitrate isotope observations.*
**Response:** Thanks for your suggestion. We have added the ice-core observed nitrate isotopes in new Figure.4

[Figure]

**Figure 4.** Sensitivity results of the modeled isotopes, i.e., $\delta^{15}N(NO_3^-)$ (a) and $\Delta^{17}O(NO_3^-)$ (b), to TCO and snow accumulation rate. Grey curve: ice core observed record; Red curve: modeled results with observed accumulation rate and TCO; Green curve: modeled results with observed TCO but mean accumulation rate throughout the record; Blue curve: modeled results with observed accumulation rate but TCO were kept the same before and after 1976. Grey shading area represents the ozone hole period.

---

## Author Response (AR2)

Dear Dr. Muller

We are grateful for the efforts of editor and reviewers to review this manuscript, and for their comments and suggestions that improve this manuscript significantly. In the response below we have addressed the remaining minor comments.

**Editor**

*Comments to the author:*

*Dear Authors,*

*congratulations!*

*I also would ask you to address the remaining minor comments (see reviews).*

*One little issue from my side:*

*l. 18 change "was of" -> "showed"*

**Response:** Thank you, this has been corrected in the revised manuscript.

*Greetings Rolf Müller*

**Anonymous Referee #1**

*The authors made a good effort to extensively consider reviewer comments. Although the prediction about Dome A's capabilities of preserving the ozone hole signals in its δ15N is somewhat challenged by the actual observation reported by Shi et al (2022), the manuscript in my opinion is still very interesting and informative. As a result, I happily recommend publication after a few points are resolved.*

*Line 57: The relative word "which" is still needed, but a comma must be inserted before "which".*

**Response:** Thank you, we have revised it accordingly.

*Line 139: Please define "UW".*

**Response:** UW refers to University of Washington. In the revised manuscript, we have spelled out this abbreviation.

*Line 231: There appears to be a missing "±" sign between 0.073 and 0.029.*

**Response:** Thank you, we have added the "±" sign in the revised manuscript.

*Line 374 and onward: It is argued that "However, at east Antarctic Plateau sites (i.e., Vostok, Dome C and Dome A) where snow accumulation rates are extremely low, δ15N(NO3-) of preserved nitrate is above 300 ‰. It would be difficult to determine changes of ~ 30 ‰ out of more than 300 ‰, ..." This might not necessarily be the case, because what matters is the variability of δ15N(NO3-). Suppose in an extreme case where the δ15N(NO3-) is simply a flat line, then the exact δ values do not matter.*
*You could instead calculate the standard deviation (1σ) of the observed, detrended δ15N(NO3-) in*

*those sites if those data are available. At least the Dome A dataset is publicly available at https://doi.org/10.11888/Cryos.tpdc.272669, kudos to the authors of Shi et al (2022). Then you could compare the 2.8\*σ (the minimal difference in order to be statistically significant at p = 0.05) with the model prediction of Δ(δ15N(NO3-)). This comparison could potentially represent a more rigorous and robust statistical analysis and actually answers if a signal could be retrieved.*

**Response:** We agree that in case the variability of $\delta^{15}N(NO_3^-)$ is very small than the response of isotope to the ozone hole can be detected. In the revised manuscript of the last submission we have emphasized this situation in lines 370 to 371 as follows: "…gradual increase in $\delta^{15}N(NO_3^-)$ might still be possibly detected as long as snow accumulation rate at these sites stayed relatively constant before and in the period of the ozone hole…".

Regarding the observed variations in $\delta^{15}N(NO_3^-)$ at Dome A and Dome C, the 2 standard deviation is above 40 ‰, i.e., Dome A ~ 40 ‰ (Shi et al., 2022) and Dome C ~ 44 ‰ (Erbland et al., 2013), without considering the values in the photic zone. These values are all larger than the modeled response of $\Delta(\delta^{15}N(NO_3^-))$ to the ozone hole. This is due to the large variations in the annual snow accumulation at these sites, and since at these sites the snow accumulation rates are very low, small changes in snow accumulation rate would lead to large enough changes in $\delta^{15}N(NO_3^-)$ that is comparable to that caused by the ozone hole.

*Line 394: "Non-local" processes?*

**Response:** Thank you, we have replaced "No local processes" with "Non-local processes" in the revised manuscript.

*Line 460 & 500-501: It has also been observed in Dome A (Shi et al, 2022). If this is a pan-Antarctic phenomenon, it might be used to strengthen your argument in Section 4.4.3.*

**Response:** Thank you for this suggestion, in the revised manuscript we have included the data from Shi et al. (2022) and made changes as follows:

In Section 4.4.3: "Similar decreases in $\Delta^{17}O(NO_3^-)$ over the past few decades were also observed in other Antarctic ice cores. Sofen et al. (2014) found that in the WAIS Divide ice core, $\Delta^{17}O(NO_3^-)$ has a long-term downward trend in the past 150 years, and a step decrease occurred after the 1970s. Meanwhile, $\delta^{15}N(NO_3^-)$ in the WAIS Divide ice core over the same period of $\Delta^{17}O(NO_3^-)$ decrease didn't have any long-term trends. A recent study by Shi et al. (2022) also indicate a downward trend of $\Delta^{17}O(NO_3^-)$ after the 1970s which is unlikely be explained by the effects of the ozone hole. These coherent decreases in $\Delta^{17}O(NO_3^-)$ in West, East and Central Antarctica after the 1970s may imply changes in nitrate chemistry in the source region. Assisted by box-model sensitivity studies, Sofen et al. (2014) have attributed the WAIS Divide ice-core $\Delta^{17}O(NO_3^-)$ decrease in the past 150 years (including that after the 1970s) to decreases in the $O_3$ to $RO_2$ ratio in extratropical Southern Hemisphere $NO_x$ source regions. Decreases in $O_3$ to $RO_2$ ratio means a reduced importance of $O_3$ oxidation in the conversion of NO to $NO_2$, leading to lower $\Delta^{17}O(NO_3^-)$ and subsequently lower $\Delta^{17}O(NO_3^-)$. Long-range transport of nitrate from the $NO_x$ source regions to Antarctica can then lead to lower $\Delta^{17}O(NO_3^-)$ in primary nitrate. This at least qualitatively explains the observed decreasing $\Delta^{17}O(NO_3^-)$ trend."

In addition, in the conclusion we have also added a statement to include the $\Delta^{17}O(NO_3^-)$ data from Shi et al. (2022):

"…Such decreases in the same period have also been observed in the WAIS Divide ice core (Sofen et al., 2014) and Dome A snow pit (Shi et al., 2022). This decrease can't be explained by post-depositional processing even including the effects of the ozone hole…"

**Anonymous Referee #2**
*The authors have done a good job responding to and addressing the reviewer comments. Thank you for this valuable contribution to the literature. I look forward to seeing the paper published.*

*Regarding the Shi et al. (2022) study at Dome A, I agree with the authors and am pleased to read their independent assessment of the study in terms of:*
*- The timing of changes in the nitrate isotope snow pit record and ozone do not line up*
*- The modelling results cannot be reproduced given the parameters reported*
*At present, the modelling and observations at Dome A do not appear to support the conclusions drawn in Shi et al. (2022). This is not to say that d15N-NO3 should be discarded if other processes are taken into account. My suggestion for the final manuscript is for the authors to emphasize the difference in the Dome A TRANSITS results between this study and those reported in Shi et al. (2022) to leave the scientific debate open about the sensitivity of the ozone proxy at the site.*

**Response:** We agree. Actually, in the revised manuscript we submitted last time, we have briefly compared the model concentration and $\Delta^{17}O(NO_3^-)$ with the observations and modeled results from Shi et al. (2022). In the final form of this submission, we have emphasized more the differences of the two TRANSITS model outputs in Section 4.3 (blue is the added statements):

"Note Shi et al. (2022) also did TRANSITS modeling study, but the model parameters are not clear, e.g., snow e-folding depth, quantum yield of snow nitrate photolysis, and the modeled results can't be reproduced given local Dome A conditions we complied. For example, Shi et al. (2022) stated when modeling the Dome A situation, similar parameters to Dome C (Erbland et al., 2015) were used except snow accumulation rate and TCO. However, the quantum yield is 0.026 (Erbland et al., 2015), and using this same quantum yield at Dome A will give a predicted $\delta^{15}N(NO_3^-)$ value of 1150 ‰ in preserved snow, three times higher than the observations as well as the modeled results of Shi et al. (2022). In addition, Shi et al. (2022) didn't present the modeled result of $\Delta^{17}O(NO_3^-)$ though the model is able to predict $\Delta^{17}O(NO_3^-)$."

Note in this manuscript we don't want to spend too much time on the results of Dome A presented by Shi et al. (2022), and our coauthors are going to have another study with measured Dome C data to explicitly deal with the Dome A case.

*What do the results in Table 1 imply for the use of d15N-NO3 as an ozone proxy?*
**Response:** The results in Table 1 indicates that the magnitude of the response of $\delta^{15}N(NO_3^-)$ to the ozone hole is larger at site with lower snow accumulation rate. Ideally, the response is more possible to be identified at sites with lower snow accumulation rate. However, whether the response can be detected also depends on many other factors (e.g., variations in snow accumulations rate, snow LAIs, etc.).

**Specific comments:**
*L112 "firn core" should be "snow pit".*

**Response:** We agree and have revised it accordingly.

*L111-116 Please proofread here and ALL additional text.*

**Response:** Thanks for this suggestion. We have proofread this sentence and all other text with necessary corrections.

*L380-381 Please add how much ozone changed over this period and include the dates. This would be useful information for Table 1 caption too. Otherwise, it is not clear what the nitrate isotope response corresponds to.*

**Response:** Thanks for this suggestion. We have added a column that indicate the spring ozone depletion value from 1979 to 1998 in Table 1 and also added the dates in this sentence:

| Site name | Latitude | Longitude | Snow accumulation Rate | Spring ozone Depletion[a] | $\Delta(\delta^{15}N(NO_3^-))$ | $\Delta(\Delta^{17}O(NO_3^-))$ |
|---|---|---|---|---|---|---|
| | ($\circ$) | ($\circ$) | (kg m$^{-2}$ yr$^{-1}$) | DU | ‰ | ‰ |
| South Pole | -90 | 0 | 75 | 158 | 6.9 | -0.8 |
| Dome A | -80.5 | 77.12 | 24.4 | 139 | 26.5 | -0.9 |
| Dome C | -75.1 | 123.33 | 28 | 171 | 30.7 | -1.1 |
| Vostok | -78.47 | 106.84 | 21.5 | 165 | 31.2 | -1.2 |
| Dome Fuji | -77.32 | 39.7 | 28.8 | 166 | 16.3 | -1.2 |
| WAIS Divide | -79.48 | -112.09 | 200 | 143 | 1.8 | -0.7 |

a. The depletion refers to the difference of the spring TCO before the ozone hole period (use that in 1979 as the representative) and that in the year of 1998 when the ozone hole was the largest.

*L398 delete "quick" and replace with "our" or similar.*

**Response:** Thanks for this suggestion. We have done this in the revised manuscript.